# Structure-function analyses reveal key molecular determinants of HIV-1 CRF01_AE resistance to the entry inhibitor temsavir

Jérémie Prévost[1,2,10], Yaozong Chen [3,10], Fei Zhou[4], William D. Tolbert [3], Romain Gasser [1,2], Halima Medjahed[1], Manon Nayrac [1,2], Dung N. Nguyen[3], Suneetha Gottumukkala[3], Ann J. Hessell [5], Venigalla B. Rao [6], Edwin Pozharski[7,8], Rick K. Huang[9], Doreen Matthies [4], Andrés Finzi[1,2] ✉ & Marzena Pazgier[3] ✉

The HIV-1 entry inhibitor temsavir prevents the viral receptor CD4 (cluster of differentiation 4) from interacting with the envelope glycoprotein (Env) and blocks its conformational changes. To do this, temsavir relies on the presence of a residue with small side chain at position 375 in Env and is unable to neutralize viral strains like CRF01_AE carrying His375. Here we investigate the mechanism of temsavir resistance and show that residue 375 is not the sole determinant of resistance. At least six additional residues within the gp120 inner domain layers, including five distant from the drug-binding pocket, contribute to resistance. A detailed structure-function analysis using engineered viruses and soluble trimer variants reveals that the molecular basis of resistance is mediated by crosstalk between His375 and the inner domain layers. Furthermore, our data confirm that temsavir can adjust its binding mode to accommodate changes in Env conformation, a property that likely contributes to its broad antiviral activity.

HIV-1 is a retrovirus that integrates its genetic information into host cells upon infection, which leads to acquired immunodeficiency syndrome (AIDS) if left untreated. Currently, there are approximately 38.4 million people worldwide that are infected with HIV-1 (UNAIDS 2021 Fact Sheet). Several types of antiretroviral small-molecule inhibitors, including protease inhibitors (PIs)[1], nucleoside and non-nucleoside reverse transcriptase inhibitors (NRTIs and NNRTIs)[2–4], integrase inhibitors (INSTIs)[5], post-attachment inhibitors[6], CCR5 antagonists[7] and fusion inhibitors[8] have been approved for HIV-1 treatment.

Nevertheless, none of the aforementioned licensed drugs specifically target the initial step of viral entry, which is initiated and mediated by the interaction of the host receptor CD4 and the envelope glycoprotein trimer (Env) expressed on the virion surface.

For many years, the attachment interface has been considered a viable target for the discovery and development of inhibitors, which are based on small molecule compounds and peptides that mimic CD4 binding to Env to interfere with CD4 engagement, with little success. Recently, the FDA has approved fostemsavir (BMS-663068;

[1]Centre de Recherche du CHUM, Montreal, QC, Canada. [2]Département de Microbiologie, Infectiologie et Immunologie, Université de Montréal, Montreal, QC, Canada. [3]Infectious Disease Division, Department of Medicine, Uniformed Services University of the Health Sciences, Bethesda, MD, USA. [4]Unit on Structural Biology, Division of Basic and Translational Biophysics, Eunice Kennedy Shriver National Institute of Child Health and Human Development, National Institutes of Health, Bethesda, MD, USA. [5]Division of Pathobiology and Immunology, Oregon National Primate Research Center, Oregon Health and Science University, Beaverton, OR, USA. [6]Department of Biology, the Catholic University of America, Washington, DC, USA. [7]Institute for Bioscience and Biotechnology Research, Rockville, MD 20850, USA. [8]Department of Biochemistry and Molecular Biology, University of Maryland School of Medicine, Baltimore, MD, USA. [9]Laboratory of Cell Biology, National Cancer Institute, National Institutes of Health, Bethesda, USA. [10]These authors contributed equally: Jérémie Prévost, Yaozong Chen. ✉e-mail: andres.finzi@umontreal.ca; marzena.pazgier@usuhs.edu

GSK3684934, Rukobia), a first-in-class attachment inhibitor for treatment in adults with multidrug-resistant HIV-1 infection. Temsavir (BMS-626529) is the active metabolite of fostemsavir and was derived from BMS-378806, first identified by a phenotypic inhibition assay in 2003 by Bristol-Myers Squibb and now marketed by ViiV Healthcare[9–11].

Structural studies involving temsavir and related analogs indicate that these compounds bind to Env within a conserved binding pocket that partially overlaps the CD4 binding site, i.e. utilizing the pocket where Phe43 of CD4 is normally inserted (referred to as the Phe43 cavity)[12–15]. However, in contrast to CD4, temsavir docks itself under the β20-β21 loop of Env in a channel orthogonal to the Phe43 cavity[13]. Currently, temsavir-like compounds are classified as conformational blockers that are capable of interfering with the CD4-induced transition of HIV-1 Env from the 'closed' State 1 to the 'open' State 3, pre-hairpin intermediate[12,16–20]. Some functional studies have also suggested that the binding of temsavir-like compounds to Env can block CD4 attachment by steric hindrance[13,21]. Site-directed, in vitro–selected, and clinical-resistant mutations have helped define temsavir's binding region within the Env gp120 subunit[13,22–28]. Computer modeling studies have been performed to explore possible binding modes with the goal of elucidating the mechanism of action for these small molecules. The recent structure of temsavir in complex with the BG505 SOSIP.664 trimer[13] helped to describe its mechanism of action at the molecular level and confirmed the capacity of temsavir to lock Env in a "closed" State 1 conformation. By binding within the interface between the nascent bridging sheet and the gp120 inner domain, temsavir stabilizes the closed Env conformation and prevents CD4-induced conformational rearrangements of the inner domain layers that lead to the formation of the bridging sheet which is necessary for coreceptor binding[13,16,18,19,29,30].

The great majority of currently circulating HIV-1 strains are susceptible to temsavir, therefore inhibitors of this type are considered as broad-spectrum drugs capable of treating multidrug-resistant HIV-1 infections[12,27,31–35]. However, as an agent that partially depends on the Phe43 cavity, temsavir relies on the size of the side chain of the residue at position 375 and has shown to be unable to effectively bind HIV-1 strains that carry residues with larger/bulkier side chains, such as the circulating recombinant form CRF01_AE that carries a naturally-occurring and highly conserved His at position 375[36–39]. We combined mutagenesis with detailed structure-function analyses to investigate the roles of obstructing/larger residues in the Phe43 cavity for the resistance of HIV-1 strains to temsavir. Our data indicate that a larger/bulkier residue at position 375 prevents effective binding of temsavir to the Envs of various HIV-1 strains and is an important but not the sole element responsible for resistance. Detailed structural analyses using a transmitted/founder CRF01_AE SOSIP.664 Env trimer with and without sensitivity-restoring mutations in complex with temsavir describe the basis of the HIV-1 drug resistance at the molecular level and reveal key differences in the modes of temsavir binding among HIV-1 strains. This information could potentially lead to the development of more potent and broadly active temsavir analogs.

## Results

### Resistance of CRF01_AE strains to temsavir is mediated by His375 and residues in the inner domain layers

To identify Env features conferring resistance to temsavir, we reanalyzed the neutralization sensitivity data previously published by Pancera et al.[13]. These data were reanalyzed with a focus on the clade or the identity of the polymorphic residues in the gp120 Phe43 cavity. Temsavir has been reported to be active against 91% of a panel of 208 HIV-1 Envs from different clades tested in this study (Fig. 1a), matching the breadth of the broadly neutralizing antibody (bnAb) VRC01[13,40]. As expected, most strains with complete resistance to temsavir belonged to the CRF01_AE subtype. This resistance has been

previously attributed to the presence of the naturally occurring residue His375, which directly interferes with the docking of temsavir-like molecules to the Phe43 cavity[36]. Most sensitive HIV-1 strains on the other hand harbor much smaller residues at this position, either Ser375 or Thr375, which can accommodate the temsavir phenyl ring (Fig. 1b). Interestingly, uncommon amino acids at residue 375 are also found in a small subset of strains, including Asn375 or Met375, which also seem to render Env more resistant to temsavir's activity. To confirm these observations, we mutated residue 375 of the clade B tier-2 JR-FL Env by a series of amino acids ranging from small (Ser and Thr) to intermediate (Asn and Met) to more bulky sidechain residues (Phe, Tyr, His, Trp). Using a single round neutralization assay, we observed that the activity of temsavir inversely correlated with the size of residue 375 (Fig. 1c). This was supported by the results of a soluble CD4 (sCD4) competition assay, where temsavir inhibited Env-CD4 interactions more efficiently in the presence of smaller 375 residues (Fig. 1d). Conversely, mutating the highly conserved His375 in CRF01_AE strains back to Ser or Thr did not restore temsavir sensitivity, suggesting that additional Env residues are involved in CRF01_AE resistance to temsavir (Fig. 2b–e).

We recently identify a set of six residues within the gp120 inner domain Layers 1, 2 and 3 (at Env position 61, 105, 108, 474, 475, and 476) that coevolved with residue 375 to facilitate interaction with CD4[38]. These residues were also shown to be involved in Env sensitivity to small molecules targeting the CD4-binding site such as CD4 mimetic compounds (CD4mc), cyclic peptide triazoles and CD4-binding site antibodies[39]. Since some of these residues are proximal to the temsavir binding pocket, we sought to determine if they also contributed to temsavir resistance. We reanalyzed the neutralization results reported by Pancera et al.[13]. and noticed that residues His61, Gln105, Val108, Asn474, Ile475, and Lys476 were all associated with higher resistance to temsavir-mediated neutralization (Fig. 2a). It is important to highlight that these changes are usually found in CRF01_AE strains, where they coevolved to accommodate the presence of His375 by reshaping the Phe43 cavity and CD4 binding site[38,39]. Therefore, we mutated these six residues in two CRF01_AE Envs (92TH023 and CM244) to harbor residues associated with higher temsavir sensitivity (His61Tyr, Gln105His, Val108Ile, Asn474Asp, Ile475Met, Lys476Arg; collectively referred to as Layer Mutations or LM). The mutations were introduced in combination with changes at position 375. The sensitivity of CRF01_AE Env to temsavir neutralization was only restored when the six-layer mutations were combined with a small residue at position 375, i.e. Ser or Thr (LMHS or LMHT) (Fig. 2b–e). Conversely when clade B Env YU2 residues were changed to CRF01_AE counterparts (Tyr61His, His105Gln, Ile108Val, Asp474Asn, Met475Ile, Arg476Lys), we observed an increased resistance to temsavir, especially in combination with the Ser375His change (LMSH), confirming the role of these layer residues in modulating the temsavir sensitivity of Env (Fig. 2f, g). Among them, layer 3 residues (474, 475 and 476) appear to contribute the most to temsavir neutralization resistance (Fig S1).

### Effect of temsavir on CRF01_AE Env conformation

Our mutagenesis and temsavir sensitivity studies confirmed the requirement of mutations of both the His375 to Ser or Thr within the Phe43 cavity and the LM residues for restoration of temsavir sensitivity to CRF01_AE. In order to understand the molecular basis of the observed cooperativity between the Phe43 cavity and the LM residues for temsavir binding, we introduced the LMHS modifications into the previously described CRF01_AE T/F100 SOSIP.664 trimer[41]. This Env was derived from a transmitted/founder CRF01_AE strain isolated from participant 40100 of the RV217 Early Capture HIV Cohort Study conducted in Thailand[42].

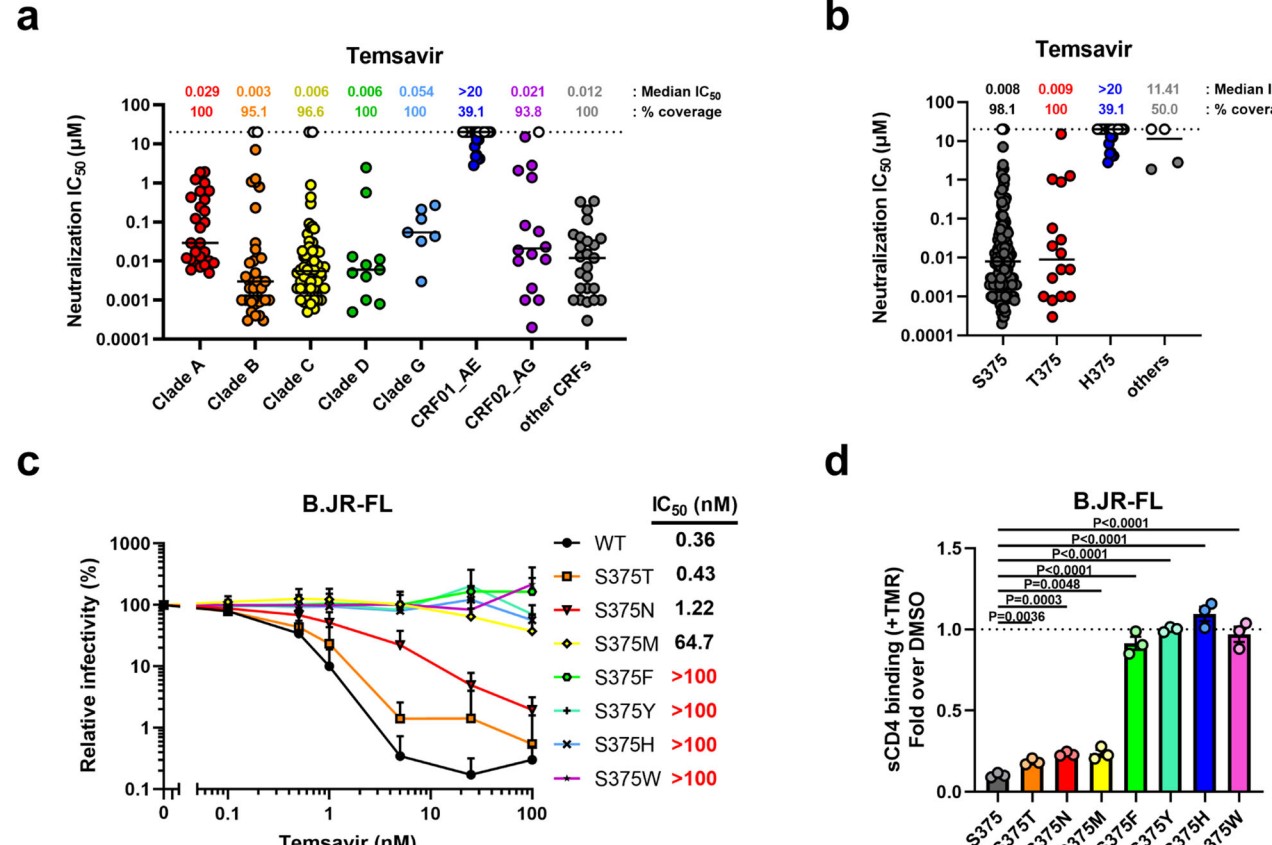

**Fig. 1 | Intrinsic resistance of HIV-1 CRF01_AE strains to neutralization by attachment inhibitor temsavir. a**, **b** The ability of temsavir to neutralize viral particles from a panel of 208 different strains was previously reported by Pancera et al.[13]. These data were reanalyzed with a focus on (**a**) the clade or (**b**) the identity of the polymorphic residue 375 in the Phe43 cavity of the gp120 subunit of Env (*n* = 208 biologically independent viral strains). Horizontal lines indicate median values. **c** Recombinant HIV-1 pseudoviruses expressing luciferase and bearing wild type (WT) or mutant Env$_{JR-FL}$ were used to infect Cf2Th-CD4/CCR5 cells in the presence of increasing concentrations of temsavir. Infectivity at each dilution of the compound tested is shown as the percentage of infection without the compound for each particular mutant. Quadruplicate samples were analyzed in each experiment. Data shown are the means of results obtained in *n* = 3 independent experiments. The error bars represent the standard deviations. Neutralization half maximal inhibitory concentration (IC$_{50}$) were calculated by non-linear regression using the Graphpad Prism software. **d** Capacity of temsavir to compete with CD4 binding as evaluated by cell-surface staining of HEK293T cells transfected with a HIV-1$_{JR-FL}$ Env expressor WT or its mutated counterpart. Binding of soluble CD4 (sCD4) in the presence of temsavir (10 μM) or an equal amount of DMSO was detected with the anti-CD4 OKT4 monoclonal antibody (mAb). Shown are the mean fluorescence intensities (MFI) obtained in the presence of temsavir normalized to the MFI in the absence of temsavir (DMSO) from the transfected (GFP+) population for staining obtained in *n* = 3 independent experiments. MFI values were normalized to the values obtained with anti-Env 2G12 mAb for each Env mutant. Error bars indicate mean ± SEM. Statistical significance was tested using a two-tailed unpaired *t*-test. Source data are provided as a Source Data file.

To characterize the structural changes induced by the LMHS mutations, we determined the Cryo-EM structures of wild type CRF01_AE T/F100 SOSIP.664 and its LMHS mutant unbound and bound to temsavir (Supplementary Figs. S2–S4, Supplementary Table S1). To allow for an unbiased analysis of the structural changes induced by the LMHS mutations and/or the binding of temsavir, we determined each structure with the same set of chaperone antibody Fabs from the 8ANC195 and 10–1074 bnAbs. In addition, to allow for a more unbiased comparison of the temsavir binding pocket formed within the LHMS CRF01_AE T/F100 to the pockets of other clades we also solved a temsavir-bound Clade A BG505 SOSIP.664 Cryo-EM structure with the same set of chaperones Fabs (Supplementary Figs. S6 and S7). The only other temsavir bound-BG505 SOSIP.664 complex was solved by X-Ray crystallography using the chaperone Fabs of PGT122 and 35022 (PDB: 5U7O[13]). With these two chaperone Fabs, we were able to get relatively high-resolution structures of the trimer including a structure of the temsavir bound-T/F100 LMHS SOSIP.664 and temsavir bound-BG505 SOSIP.664 complex to 3.1 Å and 3.0 Å resolution respectively, where the densities for temsavir were well defined enabling us to make an unambiguous placement of

the compound. In Fig. 3 we show the three structures with chaperone bound Fabs along with the changes to the overall assembly of the trimer and calculate the degree of trimer opening as defined in Tolbert et al.[43]. which describes the changes to the distances between residues at position 375 (a, b, c) and the calculated center of the gp41 portion of the trimer (d, e, f). The degree of opening is then defined relative to the closed unliganded BG505 trimer (PDB ID: 4ZMJ). The structural alignments indicate no significant changes to the overall conformation of the T/F100 LMHS SOSIP.664 mutant as compared to its wild type counterpart. We measured similar distances between residues 375 and similar degrees of opening (Fig. 3b and c). Furthermore, both structures could be superimposed with an RMSD value of 1.18 Å for the main chain atoms of the trimer. Interestingly, both the wild type T/F100 SOSIP.664 and its LMHS mutant appear to be slightly 'more open' than the Clade A, BG505 SOSIP.664 used here as a reference (the rotation/opening angles are 3.28° and 3.39° for wild type and LMHS mutant T/F100 SOSIP.664 respectively, Fig. 3b, c). This indicates a greater propensity of CRF01_AE T/F100 trimers to sample 'open' conformations, a property suggested previously to be a hallmark of CRF01_AE strains[37,39,44]. Binding of temsavir to T/F100

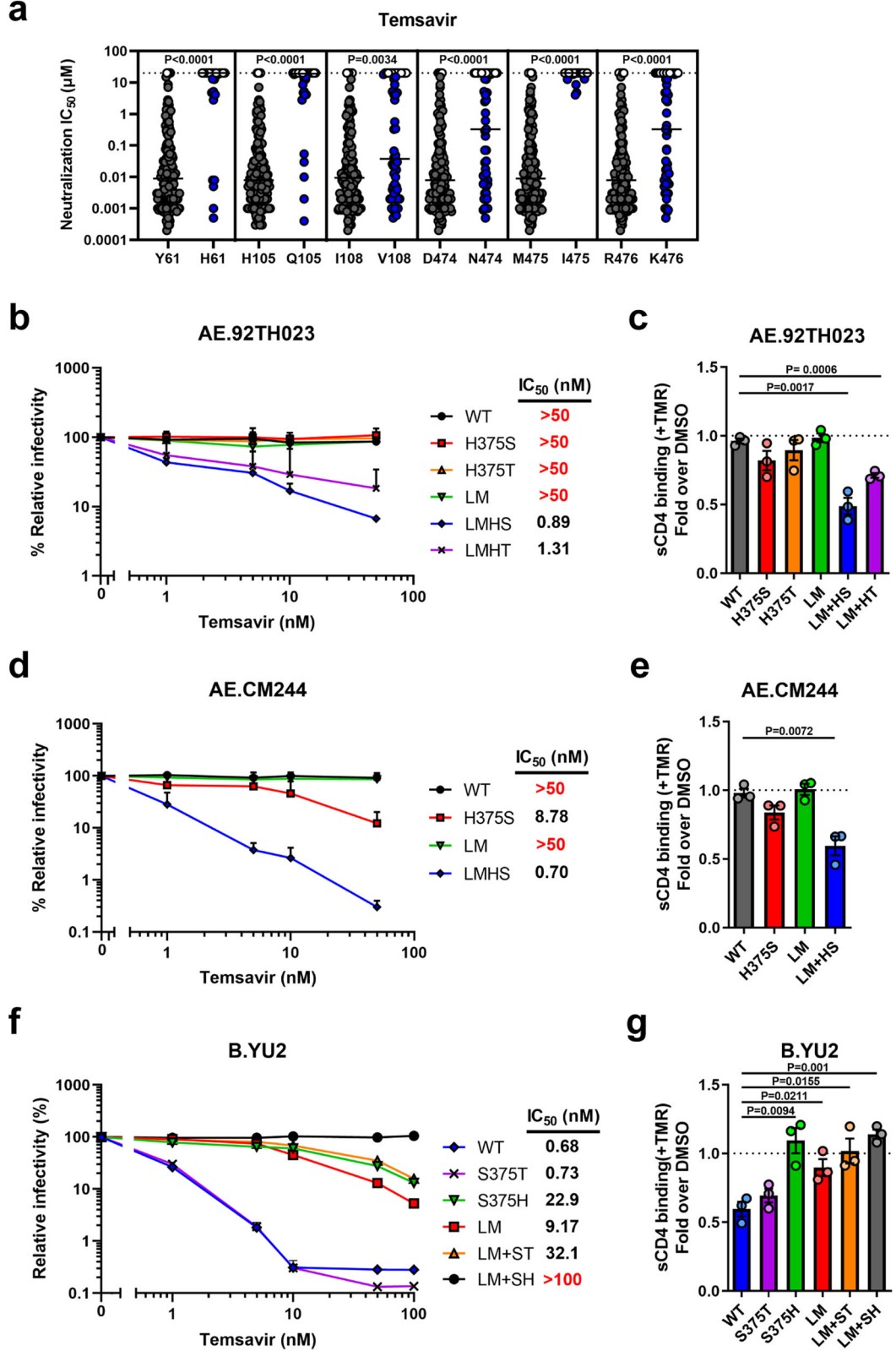

LMHS SOSIP.664 induced a further opening of the trimer as indicated by a rotation angle of 4.92° in the temsavir bound LMHS SOSIP.664. Furthermore, the temsavir- T/F100 LMHS SOSIP.664 trimer is open by 1.4° more than the temsavir-BG505 SOSIP.664 determined in this study and about 3.4° more than the temsavir-BG505 SOSIP.664 complex determined previously (PDB 5U7O, Fig. 3b, c). The differences in the angle opening between the two temsavir-BG505

SOSIP.664 complexes could possibly be attributed to the different chaperone Fabs sets used to determine both structures or to the influence of crystal packing in the latter structure determined by x-ray crystallography (Fig. 3b, c). However, the angle of opening for the temsavir bound LMHS SOSIP.664 was significantly lower than that observed for fully open, CD4-triggered Env (the angle of rotation/opening is 18.29° for the CD4 bound BG505 SOSIP.664, Fig. 3c).

**Fig. 2 | Impact of gp120 inner domain Layer residues on temsavir neutralization sensitivity. a** The ability of temsavir to neutralize viral particles from a panel of 208 Env strains was previously evaluated by Pancera et al.[13]. These data were reanalyzed with a focus on the identity of polymorphic residues (61, 105, 108, 474, 475, 476) in the inner domain of the gp120 subunit in Env (*n* = 208 biologically independent viral strains). Residues that co-evolved with Ser375 or His375 are depicted in black and blue, respectively. Horizontal lines indicate median values. Recombinant HIV-1 pseudoviruses expressing luciferase and bearing WT or mutated Env from CRF01_AE strains (**b**) 92TH023, (**d**) CM244 or (**f**) clade B YU2 were used to infect Cf2Th-CD4/CCR5 cells in the presence of increasing concentrations of temsavir. Infectivity at each dilution of the compound tested is shown as the percentage of infection without the compound for each particular mutant. Quadruplicate samples were analyzed in each experiment. Data shown are the means of results obtained in *n* = 3 independent experiments. The error bars represent the standard deviations. Neutralization half maximal inhibitory concentration (IC$_{50}$) were calculated by non-linear regression using the Graphpad Prism software. Capacity of temsavir to compete with CD4 binding as evaluated by cell-surface staining of HEK293T cells transfected with Env expressors from WT or mutated CRF01_AE strains (**c**) 92TH023, (**e**) CM244 or (**g**) clade B YU2. Binding of sCD4 in the presence of temsavir (10 μM) or equal amount of DMSO was detected with the anti-CD4 mAb OKT4. Shown are the mean fluorescence intensities (MFI) obtained in the presence of temsavir normalized to the MFI in the absence of temsavir (DMSO) from the transfected (GFP+) population for staining obtained in *n* = 3 independent experiments. MFI values were normalized to the values obtained with anti-Env 2G12 mAb for each Env mutant. Error bars indicate mean ± SEM. Statistical significance was tested using a two-tailed unpaired *t*-test. Source data are provided as a Source Data file.

We then aligned and compared the root-mean-square deviation (RMSD) values of the three CRF01_AE T/F100 trimers as follows: 1) CRF01_AE T/F100 SOSIP.664 wild type and its LMHS mutant and 2) LMHS mutant apo and its temsavir bound counterpart (Supplementary Fig. S8). We observed more pronounced differences within regions of the gp120 inner domain Layers 1, 2 and 3, the α5 helix, the β20-β21 loop and regions neighboring the fusion peptide (Supplementary Figs. S5 and S8). Interestingly, similar to the pattern observed for the opening of the trimer, the most pronounced differences in RMSD values in these regions were observed for the unliganded LMHS SOSIP.664 and its temsavir-bound counterpart (RMSD value of 1.40 Å for the whole trimer) and not wild type T/F100 SOSIP.664 and its LMHS mutant counterpart (RMSD value of 1.18 Å). While the changes observed for Layers 1, 2 and 3 map to regions where the LM mutations were introduced, the β20-β21 loop and the fusion peptide regions sit away from the mutation sites suggesting that their changes could be allosteric. In particular, the conformation of the region preceding the fusion peptide in the temsavir bound T/F100 LMHS SOSIP.664 differs significantly from the unliganded T/F100 LMHS SOSIP.664 and the wild type T/F100 SOSIP.664. Interestingly, in our structures of CRF01_AE T/F100 SOSIP.664 variants, the gp41 experimental densities of the region around the fusion peptide were not well defined. Almost half of the fusion peptide (residues 514–527) was not resolved and is missing in the three structures as compared to the structure of wild type T/F100 SOSIP.664 described in a previous study[41] (Supplementary Fig. S5). In addition, there is conformational heterogeneity among our structures in the region forming the fusion peptide proximal region (FPPR, residues 528–540) and the N-terminal region of heptad repeat 1 (HR1$_N$, residues 541–548). Furthermore, in the temsavir-bound LMHS mutant structure, part of the FPPR and HR1$_N$ regions were more exposed to solvent and sequestered within the hydrophobic core to lesser extent than in the wild type and unbound LMHS trimer (Supplementary Figs. S5 and S8a).

**The LM residues are central to the interaction network controlling movement of the β20–β21 loop**

As described previously for the Clade A BG505 trimer[13], the temsavir binding pocket is located at the interface between the gp120 inner and outer domains of the CRF01_AE T/F100 LMHS SOSIP.664 and it is gated by the mobility of β20–β21 loop of gp120. As shown in Fig. 4 the introduced LM mutations map to the α5 helix of the inner domain Layer 3 (Asn474Asp, Ile475Met and Lys476Arg), the α1 helix of the inner domain Layer 2 (Gln105His, Val108Ile), and the loop connecting β2 (minus) strand and α0 helix of the inner domain Layer 1 (His61-Tyr). While the Layer 1 His61Tyr mutation is located distal from the temsavir binding pocket and most likely has no effect on compound binding, the Layer 2 and Layer 3 mutations are required for temsavir binding pocket formation and temsavir binding. In the wild type CRF01_AE T/F100, Gln105 is in the center of a hydrogen bond network that stabilizes the mobile β20–β21 loop of gp120 in the conformation that blocks the entry to the temsavir binding pocket (Fig. 4b). There is an H-bond formed between the amide on Gln105, the backbone of Lys476 and the side chain indole of Trp427 that 'locks' Trp427 in the orientation permitting hydrophobic packing of Trp427 with surrounding residues, Trp69, Val108, Pro253, Val255 and Trp479. Therefore, Gln105 serves as a double-sided lock that holds both the β20-β21 loop and the α5 helix in place. Replacement of Gln105 to Histidine in the LMHS mutant eliminates this H-bond network and thus 'frees' the side chain indole of Trp427 which is less constrained in its hydrophobic environment permitting free movement of the β20–β21 loop. In addition, without the double-side lock (Gln105), LM mutations within the α1 helix (Layer 2) and the α5 helix (Layer 3) form a different network of H-bonds (Fig. 4b, center zoom-in window). The side chain of Arg476 (Layer 3) sits in the center of the network and makes contacts to both Glu102 (Layer 2) and Asp474 (Layer 3). These interactions stabilize interlayer contacts adjacent to the entrance of the temsavir pocket and coordinate with the Gln105His mutation in 'loosening up' Trp427 and allowing β20–β21 loop mobility. As reported previously[13], the β20–β21 loop mobility is essential for the formation of the temsavir binding pocket. Indeed, in the temsavir bound LMHS complex, the conformation of the β20–β21 loop changes significantly as compared to the unliganded LMHS structure with the loop needing to move away to accommodate temsavir (Fig. 4b). β20–β21 loop movement can be measured by the changes in the position of Trp427 at the entry of the temsavir binding pocket, 3.1 Å and 5.7 Å as calculated between the Cα and Cγ positions for the unliganded and bound complexes respectively. Interestingly, in the temsavir bound LMHS complex, the β20–β21 loop caps the temsavir in its pocket from the top and Trp427 directly lines the pocket with its side chain indole making contacts to the compound's piperazine group to lock the compound in its bound conformation. Altogether, our data indicate that the LM mutations 'pre-shape' HIV-1 Env to accommodate temsavir in its binding pocket, directly contributing to the conformation of the Env trimer by stabilizing the inner domain interface between Layers 2 and 3 in the area proximal to the binding pocket and gating β20–β21 loop movement.

**The temsavir binding pocket within the CRF01_AE T/F100 LMHS Env involves both conserved and new elements of the gp120 inner and outer domains**

The overall properties of the temsavir binding pocket in T/F100 LMHS SOSIP.664 and the BG505 SOSIP.664 are shown in Fig. 5. The α1, α3, and α5 helices and the β16 sheet border the pocket and the β20–β21 loop "gates" the pocket in both structures. In addition, the same set of twenty-four gp120 residues line the pocket in both structures (as listed in Fig. 5c i.e. gp120 residues 108-109, 112-113, 116-117, 200, 202, 255-256, 370, 375–377, 382, 384, 424–427, 432–434, and 475). Only two of these residues constitute substitutions

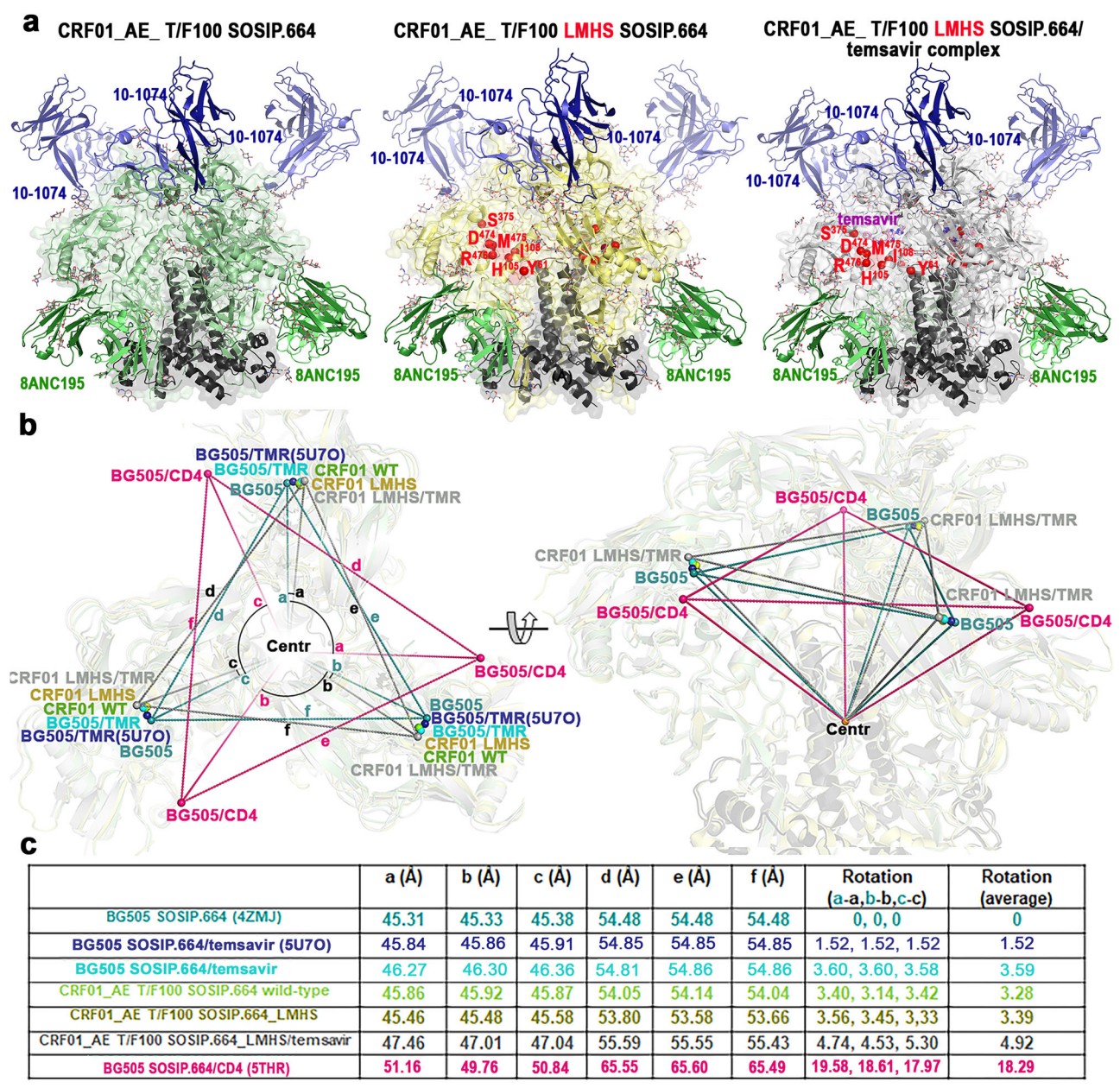

**Fig. 3 | Structure of an engineered CRF01_AE SOSIP.664 Env trimer in complex with temsavir. a** Overall structures of trimers complexed with chaperone Fabs of 10–1074 and 8ANC195 antibodies shown as cartoon with LMHS mutations highlighted in red within one gp120 promoter. Envelope sugars are shown as gray sticks. **b** Changes to overall trimer assembly, calculated as changes in position of gp120 relative to gp41 (referred to as trimer 'opening') of CRF01_AE_ T/F100 SOSIP.664 variants as compared to BG505 SOSIP.664 (PDB ID: 4ZMJ[77]). The relative position for each gp120 in the trimer is calculated based on the α-carbon position for residue 375 at the base of the CD4 Phe43 binding pocket (shown as colored spheres for each structure) relative to the gp41 trimer center (gray sphere, Centr, calculated for

all trimers aligned based on the α-carbon positions of the central gp41 α7 helices). The distances between Centr and the 375Cα of each protomer (**a–c**) and the 375Cα atoms of neighboring protomers (**d–f**) are shown to indicate the extent of the protomer rearrangement relative to gp41. The clockwise rotations of the gp120 subunits are calculated as angles relative to apo BG505 SOSIP (shown and labeled a-a', b-b', and c-c'). The BG505 SOSIP.664 bound to CD4 (PDB: 5THR[78]) is shown as a reference to the 'open' CD4-triggered conformation of trimer. c Table summarizing **a–f** distances and a-a', b-b' and c-c' rotation angles for CRF01_AE_ T/F100 SOSIP.664 complex variants relative to the unbound BG505 SOSIP trimer (PDB: 4ZMJ[77]).

introduced by the LMHS mutations in T/F100 SOSIP, His375Ser and Ile475Met (Fig. 5b). Structural superimpositions with the wild type T/F100 SOSIP unequivocally confirm that the large side chain of His375 represents the major obstacle in the temsavir binding by causing a direct steric clash to the main anchoring region of the compound 'head' phenyl moiety. The substitution of His375 with a small side chain residue, serine, mitigates these clashes and is required for the binding of the compound (the BSA of Ser375 alone in the temsavir T/F100 LMHS SOSIP.664 complex is around 9 Å). Furthermore, the

LM mutation Ile475Met introduces an important hydrophobic contact to temsavir; Met475 alone contributes more than 20 Å² of BSA to the compound-SOSIP interface. Ile475 is one of the most resistant substitutions in native T/F100 Env, reportedly accounting for ~90 fold reduction in temsavir activity[13], and has been confirmed clinically as a major contributor to reduced temsavir susceptibility for the CRF01_AE subtype[25]. As with residues lining the temsavir binding pocket, only one residue in the β3 sheet at position 202 differs between the T/F100 LMHS and BG505 SOSIP.664 (a lysine versus

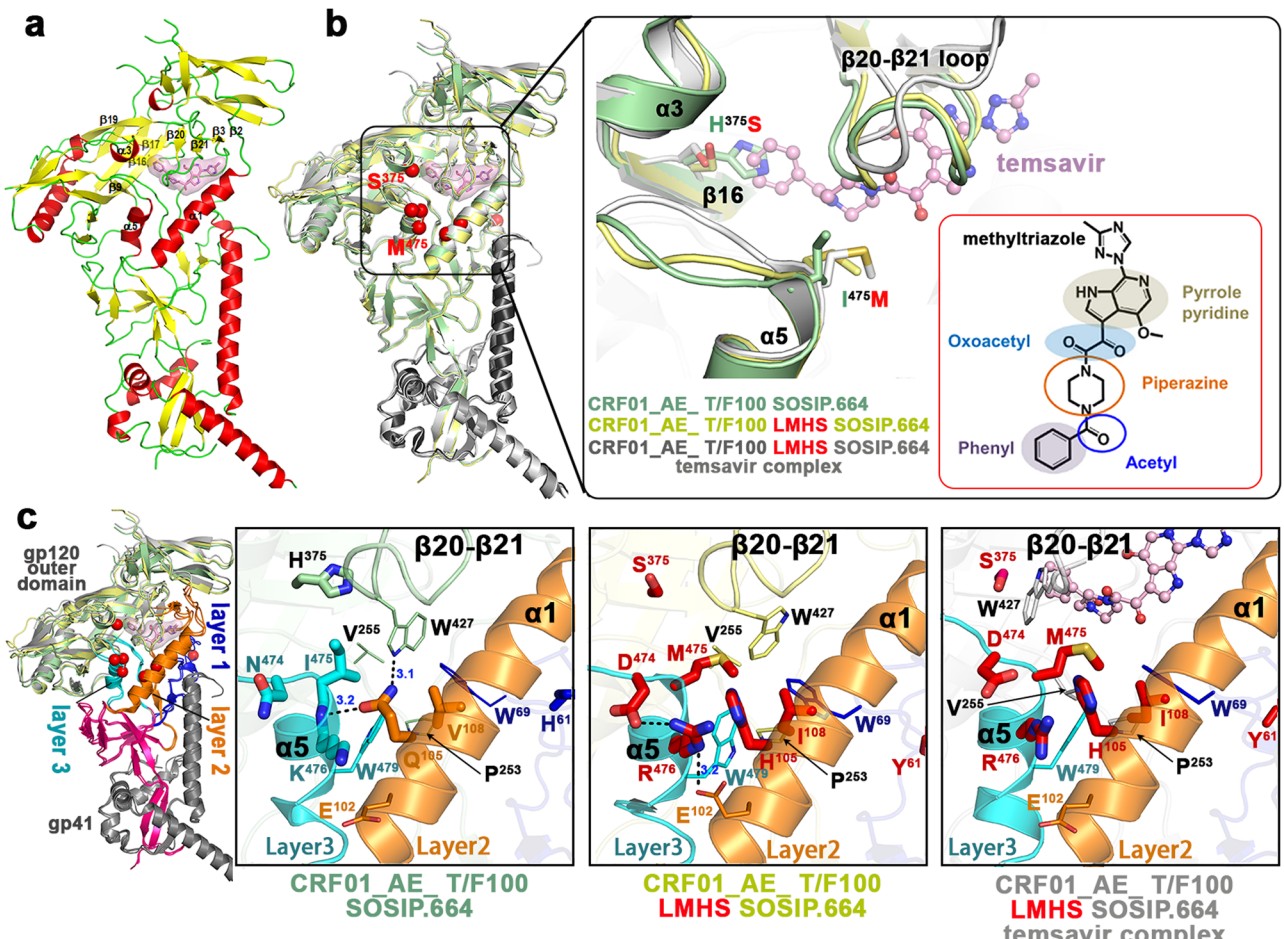

**Fig. 4 | The LMHS mutations' induced changes to the temsavir binding pocket.**
**a** Insights into regions forming the temsavir binding pocket. The gp120/gp41 protomer of temsavir-SOSIP.664 LMHS complex with secondary elements colored yellow, red and green for β-strands, α-helices and loops respectively. Temsavir is shown as stick/surface representation and secondary elements forming or surrounding the pocket are as labeled. **b** Superimposition of the CRF01_AE_ T/F100 SOSIP.664 wild type (green), CRF01_AE_ T/F100 SOSIP.664 LMHS mutant (yellow) and its complex with temsavir (gray) with a blow-up view into the β20-β21 loop region. LMHS mutations introduced in this region are shown as red spheres/

colored red. The inlet shows the chemical structure of temsavir. **c** Changes induced to the temsavir binding pocket by LMHS mutations. Inner domain Layers are colored blue, orange and cyan for Layer 1, 2 and 3 respectively, with the LMHS mutations shown within one gp120/gp41 promoter. The 7-stranded β-sandwich and N- and C-termini of the gp120 inner domain are colored magenta. The blow-up views show the network of interactions mediated by LMHS residues at the 'entry' of the temsavir binding pocket and the neighboring β20-β21 loop. Hydrogen bonds are shown as dashed blue lines.

threonine). This residue caps the binding pocket in the region that accommodates the 'tail' methyltriazole ring.

The conformations of temsavir in the T/F100 LMHS and BG505 binding pockets resemble each other relatively well with an almost identical conformation and position of the core region, i.e. the piperazine-oxoacetyl-methyl pyrrolo-pyridine moiety, and more noticeable differences in the conformations of the 'head' benzoyl moiety and the 'tail' methyltriazole ring (Fig. 5b). In the core region, two hydrogen bonds are formed between pocket residues Asp113 and Trp427 to the pyridine and the oxoacetyl group of temsavir, respectively, and are present in both structures (Fig. 5a). In addition, the core is stabilized in both complexes by hydrophobic contacts from Met426, Trp112, Ile424 and Met434 (Fig. 5). Whereas the temsavir core regions largely overlap, the 'tail' methyltriazole ring is flipped when bound to T/F100 SOSIP as compared to BG505 SOSIP, which places the methyl group oriented towards the β13 sheet instead of the C-terminus of the α1 helix (Fig. 5b, zoom-in view). This different orientation is most likely a result of the Lys202 side chain which is the only pocket residue that is different between T/F100 LMHS and BG505. In BG505, the threonine side chain at position 202 blocks the path for the extension of the BMS-626529 'tail' toward the

β13 sheet while a larger but more flexible lysine side chain allows the methyltriazole ring to adopt conformation with its methyl group 'up'. Interestingly, two recently reported temsavir derivatives, BMS-814508 and BMS-818251, which show 4- and -100-fold greater inhibition than temsavir against a laboratory-adapted NL4-3 HIV-1 strain, respectively, have a thiazole ring and longer hydrophilic tails in the place of the methyltriazole in temsavir[14]. Both temsavir-analogs-bound BG505 SOSIP structures (PDB: 6MU6 and 6MU7) demonstrate that the extended tails of both inhibitors orient toward the α1 helix instead of the β13 sheet. Given that BMS-818251 has also been reported to have improved inhibition against CRF01_AE strains with IC$_{50}$ values ranging from 32–733 nM as compared to IC$_{50}$ > 5.8 μM for temsavir[14], it would be worthwhile to determine the conformation of the extended C-terminal 'tail' that BMS-818251 adopts when bound to T/F100 LMHS to see if Lys202 plays the role of 'gate keeper' for BMS-818251 and forces different conformations for the tail in this region.

The noticeable differences in the conformation of the temsavir 'tail' region are coupled with changes to the orientation of the 'head' phenyl moiety in the T/F100 LMHS SOSIP.664 complex as compared to the temsavir-bound BG505 SOSIP.664 (Fig. 5b). In the T/F100

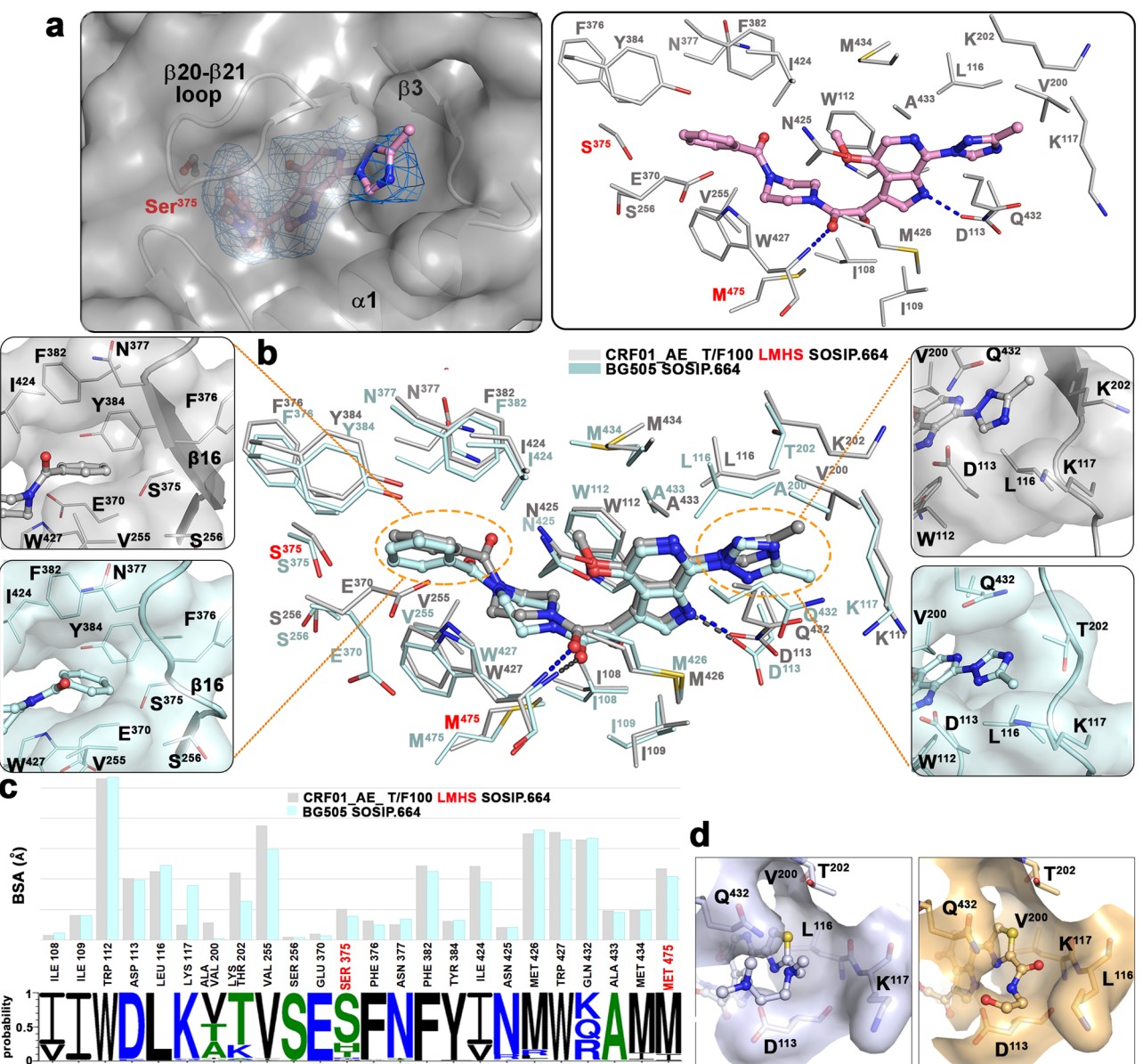

**Fig. 5 | Temsavir binding pocket. a** Temsavir in its binding pocket within the CRF01_AE_ T/F100 LMHS SOSIP.664. Cryo-EM density map of temsavir is shown (left panel) and residues lining the pocket shown as sticks (right panel). The LMHS introduced residues are labeled in red and H-bonds as blue dashes.
**b** Superimposition of the temsavir binding pockets formed within CRF01_AE_ T/F100 LMHS SOSIP.664 (gray) and BG505-SOSIP.664. The inhibitor molecules are shown in ball-and-sticks while the pocket residues are shown in sticks with or without surface. The LMHS mutations within the pockets are highlighted in red. To the right and left are close-up views into the part of the pocket accommodating the acetyl-phenyl and methyltriazole moiety of the inhibitor, respectively. **c** The

residue-resolved buried-surface-area (BSA) of gp120 contributing to the temsavir-protein interface, as determined by PISA. BSA values represent the average of the three copies in the trimer. The conservation of residues lining the temsavir pocket is shown at the bottom. The height of the residue at each position is proportional to its frequency of distribution among the HIV-1 isolates, as deposited in the Los Alamos HIV database (all clades are included). Residues are colored according to hydrophobicity: black - hydrophobic, green - neutral, blue - hydrophilic. **d** Close-up views of the extended tails on the thiazole ring from the two temsavir analogues whose structures have been determined (PDB: 6MU6 and 6MU7)[14].

LMHS complex, the phenyl ring of temsavir is tilted approximately 0.8–1.0 Å toward one edge of the pocket and packs more tightly with Phe382, Tyr384 and Ile424. We speculate that the shift of the phenyl ring in the pocket results from the side chain movements of Val255, Phe382 and Trp427 and the backbone rearrangement of the β16 sheet as compared to the conformation of this region in BG505 SOSIP. As a result, there is a strong, approximately 4 Å, face-to-edge π-π interaction between Phe382 and the temsavir phenyl ring. Concomitantly, the backbone movement of the β16 sheet is likely required to allow temsavir to bind to T/F100 LMHS SOSIP.664 and accommodate its phenyl ring. Interestingly, superimposition of the

temsavir-bound T/F100 SOSIP.664 with BMS-814508 and BMS-818251 bound BG505 SOSIP.664 suggest that replacement of the acetyl group by a nitrile in the latter two temsavir derivatives does not result in a shift of the temsavir phenyl ring. Altogether, this data indicates that the binding mode of temsavir to Env may differ among different HIV-1 strains, with polymorphic Env residues and Env conformational flexibility forcing changes in the orientation of both 'head and tail' regions of temsavir. These variables should be considered in the structure-based development of more potent and broader temsavir-like attachment inhibitors, as has been previously done with other classes of HIV-1 inhibitors[45–51].

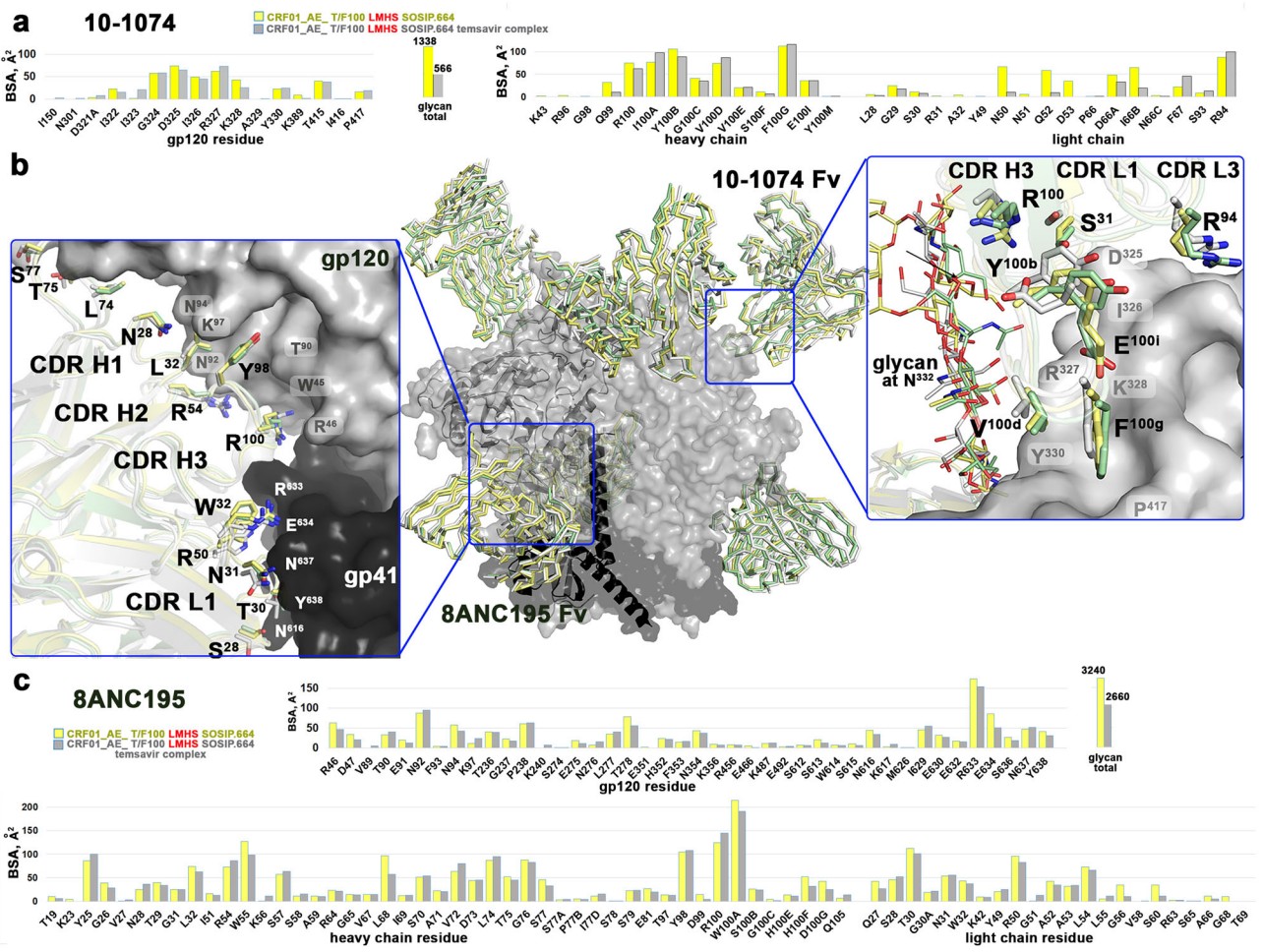

**Fig. 6 | Comparisons of the 10–1074 and 8ANC195 binding interfaces formed with CRF01_AE_ T/F100 SOSIP.664 wild type and its LMHS mutant unbound or bound to temsavir. a, c** Buried Surface Area (BSA) contributed by individual residues of Env and the Fabs of 10–1074 (**c**) and 8ANC195 (**b**) in apo CRF01_AE_ T/F100 LMHS SOSIP.664 and CRF01_AE_ T/F100 LMHS SOSIP.664 bound to temsavir. The total Env glycan contribution to the interface is shown as a separate bar with the value of BSA shown at the top. BSA values represent the average of the three copies in the trimer. **b** Superimposition of the complexes with 10–1074 and 8ANC195 colored green, yellow and gray for wild-type CRF01_AE_ T/F100 SOSIP.664, and its apo and temsavir-bound LMHS mutant complexes, respectively. Blow-ups show similarities/differences between the complexes of how individual Fab/antibody residues interact with the Env antigen.

Despite the observed conformational differences of temsavir within the T/F100 LMHS SOSIP.664 and BG505 SOSIP.664 binding cavities, the Env residues contributing to temsavir binding are highly conserved. Using residue BSA to stratify residue contribution to the interface, temsavir mostly relies on interactions to Env residues that are strictly conserved in over 99% of HIV-1 sequences (residues Trp112, Leu116, Phe382, and Trp427, Fig. 5c), highly conserved with approximately 97% or more of HIV-1 sequences (residues Asp113, Lys117, and Val255) or conserved with limited sequence variation (residues Thr/Lys202, Ile/Val424, Met/Arg/Leu426, Lys/Gln/Arg432 and Met/Ile475). The high degree of conservation implies that Env is only able to tolerate limited sequence diversity at these positions. Temsavir can accommodate many of these changes. The Thr202Lys change can be accommodated by a rearrangement of the methyltriazole ring in temsavir as mentioned previously. Asp113 which is involved in one of the two hydrogen bonds with temsavir is rarely any other residue, but the infrequently seen Glu113 or Asn113 can also form a hydrogen bond with temsavir. The residue at position 424 contributes to the binding pocket through van der Waals interactions that can be made equally well by different residues (i.e. Ile or Val). Other key interactions with temsavir are dependent on main chain atoms such as Lys/Gln/Arg432 or the hydrogen bond between Trp427 and temsavir. One of the few temsavir pocket residue sequence changes that can influence temsavir

binding is His at position 375. Temsavir resistance in many other cases is likely to come from conformational changes induced by residue changes outside of the binding pocket.

### Temsavir induced changes to the bnAb binding interface

We obtained structures of CRF01_AE_ T/F100 SOSIP.664 variants with two chaperone bnAbs: 8ANC195 and 10–1074. It was suggested previously that temsavir can synergize with the CD4 binding site (CD4bs)-targeting bnAbs in neutralizing HIV-1 strains including strains with resistance conferring mutations[52]. However, a recent study failed to provide evidence of synergy among these bnAbs and temsavir. Temsavir treatment of Env expressing cells significantly decreased the binding of most bnAbs[53].

Our structural studies of CRF01_AE_ T/F100 LMHS SOSIP.664 with or without temsavir with the same set of 8ANC195 and 10–1074 Fabs permitted us to evaluate if there were differences in the bnAb–Env binding interface induced by temsavir. Since all structures were solved at similar overall resolution, direct comparisons among complexes were possible for interfaces' details and BSA. Figure 6 shows the molecular details of the bnAb interface of 8ANC195 and 10–1074 bound to CRF01_AE_ T/F100 SOSIP.664 variants. While total BSA for 8ANC195 and 10–1074 interfaces for wild type CRF01_AE_ T/F100 SOSIP.644 are 5915 Å² and 1848 Å², respectively, they change

to 5920 Å² and 2171 Å² for the unbound LMHS SOSIP and 5536 Å² and 1680 Å² for the temsavir-bound LMHS SOSIP (Supplementary Table S2). In addition, although specific Fab-Env contacts remain largely unchanged in both complexes (Fig. 6b), the interfaces for the bnAbs form with smaller BSA for both Env antigen and antibody residues (Fig. 6a, c, Supplementary Table S2). When 10−1074 interacts with temsavir-bound T/F100 LMHS SOSIP.644, the overall contribution of the gp120 Asn332 glycan to the interface is significantly lower than in the unbound LMHS SOSIP.644 complex (BSA of 459 Å² as compared to 728 Å²). In addition, 10−1074 depends significantly less on the light chain contacts in the temsavir-bound T/F100 LMHS SOSIP.644 as compared to the unbound one (Supplementary Table S2). Overall, as described before[13], the 10−1074 footprint on Env is very small, and the main interaction is mediated by the extended CDR H3 of this bnAb. Differences in the 8ANC195 interface BSA between the temsavir-bound and unbound complexes is less pronounced but still noticeable. 8ANC195 recognizes an epitope at the gp120-gp41 junction (Fig. 6b, c), splitting the interface BSA between both Env protomers with a larger contribution to the gp41 subunit. 8ANC195, similar to 10−1074, relies less on binding to temsavir-bound T/F100 LMHS SOSIP.644 glycans as compared to unbound SOSIP (BSA of 1774 Å² as compared to 1844 Å²) and forms a complex with significantly lower BSA of heavy and light chain residues (2579 Å² as compared 2761 Å², Supplementary Table S2). Although the exact mechanism causing the differences in the bNAb complex BSA between temsavir-bound and unbound T/F100 LMHS SOSIP.644 remains difficult to pinpoint, it can be hypothesized that these differences are induced by changes in the overall trimer conformation upon temsavir binding in the CRF01_AE Env. Whether temsavir-induced changes to the overall stability of bnAb-Env complexes hold for other HIV-1 clades remains to be shown. Furthermore, the modulation of bnAb binding by temsavir is most likely to be epitope dependent.

Finally, we noticed changes to the mode of binding of the V3 glycan-targeting 10−1074 bnAb to the CRF01_AE_ T/F100 SOSIP.664 as compared to Clade A BG505 (Supplementary Fig. S9). 10−1074 locks the Env trimer in a CD4-incompetent closed conformation but the only available 10−1074 bnAb-Env complex structure is of 10−1074 Fab bound to BG505 SOSIP.664[13]. Structural comparisons indicate that 10−1074 interacts with T/F100 SOSIP.664 in a different manner compared to the BG505 SOSIP.664, despite the majority of interacting CDRH3 residues adopting a similar backbone trace in both SOSIP.664 s. As an example, Glu100I in CDRH3 (Kabat residue numbering[54]) forms two polar interactions with Arg327 and Gln328 of BG505 gp120, while this same glutamate side chain points outward without making any close contact with the V3 residues in the in the T/F100 SOSIP.664 structure. Instead, Tyr100B forms a sole hydrogen bond with Arg327 between the 10−1074 heavy chain and T/F100 SOSIP.664. Strikingly, unlike the V1 loop of BG505-SOSIP.664 that engages closely with the 10−1074 light chain, the T/F100 Env V1 loop with 10 more residue-insertions as compared to BG505 remains largely disordered and barely interacts with the light chain of 10−1074. Although high anisotropy limits the local resolution at the 10−1074-gp120 interface, the Asn332 glycan is well defined, and it is interactions with this carbohydrate that are major determinants for binding for PGT121-like and 10−1074-like antibodies[55].

## Discussion

It has been shown that the type of residue at position 375 within the Phe43 cavity modulates Env conformation with residues filling the Phe43 cavity predisposing Env to more "open" conformations[37,39,56,57]. The nature of the residue at position 375 also modulates Env sensitivity to small molecule compounds and peptide-based inhibitors that mimic CD4 (CD4mcs)[13,25,36,39,58]. Residues with larger/bulkier side chains including histidine (His375) or tryptophan (Trp375) have been shown

to obstruct the binding of CD4mc[37,39,57,59]. Temsavir only partially overlaps with CD4 and CD4mc binding sites, anchoring its phenyl group into the Phe43 cavity but also relying on other Env regions, mostly adjacent to the β20-β21 loop. Therefore, in its mechanism of action, temsavir differs from most CD4mcs by locking Env in a "closed" State 1 conformation instead of enabling Env to adopt more open conformations[30].

In spite of its large breadth, certain HIV-1 strains such as clade AE, CRF01_AE, with an invariant His375, exhibit natural resistance to temsavir. His375 blocks temsavir's interaction with the Phe43 cavity, however there is more than the nature of the residue at position 375 that determines temsavir resistance. Our studies have shown that the mechanism of resistance extends beyond steric clashes within the binding pocket. Replacement of His375 in CRF01_AE viruses with a smaller polar Ser375, naturally present in Envs from most HIV-1 clades, does not restore even partial temsavir sensitivity. The susceptibility of CRF01_AE Envs to temsavir can only be restored when the Ser375 change is combined with the alteration of six co-evolving residues within the gp120 inner domain layers that appear to act synergistically to facilitate temsavir binding.

At first glance, introduction of the combination of His375Ser and the Layer Mutations (LMHS) to CRF01_AE Env has no effect on Env's conformation or its assembly. A comparison of the structures of the wild type and the LMHS variant of T/F100 SOSIP.644 indicates that the addition of the LMHS mutations causes no significant differences to the overall trimer conformation. These structures also confirm that the CRF01_AE trimer is a slightly 'more open' than Clade A BG505 (the gp120 trimer rotation/opening angles are about 2° larger than in BG505 SOSIP.664). This is consistent with previous findings suggesting that CRF01_AE Env is prone to adopt more open conformations, a feature most likely due to the larger side chain of His375 within the Phe43 cavity[37,39,44]. However, both the wild type and the LMHS variant of T/F100 SOSIP.644 have similar opening angles which suggests either that His375 is not the sole difference responsible for opening the trimer or that the combination of His375Ser with the LM mutations enables Env to adopt a conformation similar to unmutated Env. Further studies are required, including testing of variants with single His375Ser/Thr substitutions, to fully understand the role of the residue at position 375 in the conformation of CFR01 AE Env. Temsavir binding opens T/F100 LMHS SOSIP.644 further, but it locks the Env in an intermediate 'closed' state (with an opening angle of 4.9° as compared to 18.3° for the fully CD4-triggered, 'open' trimer, PDB ID 5THR, Fig. 3). Interestingly the degree of opening induced by temsavir is significantly larger for CRF01_AE than for Clade A BG505, roughly 4.9° as compared to 3.6°. At this point, it is important to note that there is a significant difference between the opening angles for the temsavir bound-BG505 SOSIP.664 trimers whose structures were determined by different methods. The opening angle determined for the complex in this paper is 3.6° as compared to 1.5° for the trimer determined by x-ray crystallography. As mentioned previously this could be due to the use of different chaperone antibodies in structure determination or to the influence of crystal packing in the latter structure. Altogether, our data indicates that although the LMHS mutations are important to enable the formation of the nascent temsavir binding pocket, on their own, they do not have much of an impact on the global trimer conformation.

Detailed analyses of the interaction network mediated by His/ Ser375 and LM residues reveal their cooperative roles in the formation of the temsavir binding pocket and in enabling temsavir binding. In the unmutated CRF01_AE Env, Trp427 and the residues surrounding the pocket are rigidified by interactions mediated by Layers 2 and 3, the LM residues blocking entry to the pocket, and the movement of the β20-β21 loop. β20-β21 loop movement is required for pocket formation and to anchor temsavir within the pocket. Interestingly, the binding of temsavir to Env is not possible when only His375Ser or the

LM mutations alone are introduced to CRF01_AE Env, which strongly indicates intramolecular cooperativity/crosstalk between these two distal Env regions for temsavir pocket formation and the mechanism of CRF01_AE resistance to temsavir. This also confirms that there is an underestimated role for the inner domain layers in shaping the conformation of Env, not only for the binding of CD4 and CD4mcs but also for compounds like temsavir that stabilize Env in a closed state.

The temsavir binding pocket within the T/F100 LMHS SOSIP.644 resembles the temsavir binding pocket in BG505 SOSIP.664 with major differences in how the phenyl and the methyltriazole ring are anchored within the pocket. Our data indicates that the methyltriazole ring may adopt conformations in which the methyl group can be directed either 'up' towards the gp120 β13 strand or 'down' towards gp120 α1 helix. The Lys202Thr residue change between CRF01_AE T/F100 and Clade A BG505 in the area that accommodates the methyltriazole ring enables temsavir to bind in a conformation that potentially permits additions to the methyl group to bind to regions around the β13 strand. Such extensions could improve the affinity of temsavir for CRF01_AE strains, possibly overcoming the His375 and LM hurdle to binding. Our data also confirm the propensity of temsavir to slightly adjust its conformation in order to compensate for the shape and chemical environment in the pocket induced by minor changes in the conformation of Env. This could in part explain the extraordinary breadth and potent reactivity of the compound in targeting HIV-1 strains of multiple clades.

## Methods

### Cell lines
HEK293T human embryonic kidney cells and Cf2Th canine thymocytes (obtained from ATCC, Catalog# CRL-3216) were maintained at 37 °C under 5% $CO_2$ in Dulbecco's Modified Eagle Medium (DMEM) (Wisent), supplemented with 5% fetal bovine serum (FBS) (VWR) and 100 U/mL penicillin/streptomycin (Wisent). Cf2Th cells stably expressing human CD4 and CCR5 (Cf2Th-CD4/CCR5, obtained from Dr Joseph Sodroski[60]) were grown in medium supplemented with 0.4 mg/mL of G418 sulfate (Thermo Fisher Scientific) and 0.2 mg/mL of hygromycin B (Roche Diagnostics).

### Plasmids
The plasmids expressing the CRF01_AE Envs HIV-1$_{92TH023}$ and HIV-1$_{CM244}$ were previously reported in[38,61]. The plasmid pSVIIIenv expressing the clade B HIV-1$_{YU2}$ Env and the Tat-expressing plasmid (pLTR-Tat) were previously reported in[56]. The sequence of clade B HIV-1$_{JR-FL}$ Env[62] was codon optimized (GenScript) and cloned into expression plasmid pcDNA3.1(-) (Invitrogen). The plasmid encoding the transmitted/founder CRF01_AE HIV-1$_{40100}$ (T/F100) SOSIP.664 gp140 trimer was previously described in[41]. A set of modifications were introduced into the T/F100 Env sequence to generate the LMHS SOSIP.664 gp140 version, including the seven LMHS mutations (H61Y, Q105H, V108I, N474D, I475M, and K476R)[38], the three SOSIP mutations (A501C, T605C, and I599P)[63] and the replacement of its natural furin cleavage site (REKR) with an enhanced cleavage site (RRRRRR [R6]). The T/F100 SOSIP.664 gp140 is flanked by an N-terminal CD5 leader peptide and a C-terminal Twin-Strep-Tag. Mutations were introduced individually or in combination into the different Env expressors using the QuikChange II XL site-directed mutagenesis protocol (Stratagene). The presence of the desired mutations was determined by automated DNA sequencing. The numbering of all the Env amino acid sequence is based on the prototypic HXB2 strain of HIV-1, where 1 is the initial methionine[64]. A list of primer sequences used for site-directed mutagenesis is provided in Supplementary Table S3.

### Small molecule
The HIV-1 attachment inhibitor temsavir (BMS-626529) was purchased from APExBIO. The compound was dissolved in dimethyl sulfoxide (DMSO) at a stock concentration of 10 mM, aliquoted, and stored at −80 °C until further use.

### Viral neutralization assay
Cf2Th-CD4/CCR5 cells were infected with single-round luciferase-expressing HIV-1 pseudoparticles[56]. Briefly, HEK293T cells were transfected by the calcium phosphate method with the proviral vector pNL4.3 (Vpr-/Env-)Luc (NIH AIDS Reagent Program) and a plasmid expressing wild type or mutant HIV-1 Env at a ratio of 2:1. Two days after transfection, the cell supernatants were harvested. The reverse transcriptase activities of all virus preparations were measured as described previously[65]. Each virus preparation was frozen and stored in aliquots at −80 °C until further use. Twenty-four hours before the infection, Cf2Th-CD4/CCR5 target cells were seeded at a density of $1 \times 10^4$ cells/well in 96-well luminometer-compatible tissue culture plates (Corning). Luciferase-expressing recombinant viruses (10,000 reverse transcriptase units) in a final volume of 100 μL were incubated with the indicated amounts of temsavir for 1 h at 37 °C and were then added to the target cells followed by incubation for 48 h at 37 °C; the medium was then removed from each well, and the cells were lysed by the addition of 30 μL of passive lysis buffer (Promega) followed by one freeze-thaw cycle. An LB 942 TriStar luminometer (Berthold Technologies) was used to measure the luciferase activity of each well after the addition of 100 μL of luciferin buffer (15 mM $MgSO_4$, 15 mM $KH_2PO_4$ [pH 7.8]), 1 mM ATP, and 1 mM dithiothreitol) and 50 μL of 1 mM D-Luciferin free acid (Prolume). The neutralization half-maximal inhibitory concentration ($IC_{50}$) represents the amount of temsavir needed to inhibit 50% of the infection of Cf2Th-CD4/CCR5 cells by recombinant luciferase-expressing HIV-1 bearing the indicated Env.

### Soluble CD4 (sCD4) competition assay
Binding of sCD4 to cell surface Env was performed as previously described in[39]. Briefly, $2 \times 10^6$ HEK293T cells were transfected with 7 μg of Env expressor and 1 μg of a green fluorescent protein (GFP) expressor (pIRES2-EGFP; Clontech) with the calcium phosphate method. When the pSVIII Env expressor was used, it was co-transfected with 0.25 μg of a Tat-expressing plasmid. At 48 h post-transfection, HEK293T cells were detached and washed with PBS. Then, cells were incubated with 10 μg/mL of soluble CD4 (sCD4) in presence of temsavir (10 μM) or an equivalent volume of the vehicle (DMSO), followed by staining performed with the monoclonal anti-CD4 OKT4 antibody (Thermo Fisher Scientific, Catalog# 14-0048-82, 0.5 μg/mL) and goat anti-mouse IgG antibodies pre-coupled to Alexa Fluor 647 (Thermo Fisher Scientific Catalog# A21235, 2 μg/mL) to detect cell-bound sCD4. Alternatively, transfected HEK293T cells were stained with the anti-Env 2G12 antibody (NIH AIDS Reagent Program, Catalog#: ARP-1476, 10 μg/mL) and goat anti-human IgG antibodies pre-coupled to Alexa Fluor 647 (Thermo Fisher Scientific, Catalog# A21445, 2 μg/mL) to normalize the level of Env expression from each mutant. Stained cells were fixed with a PBS solution containing 2% formaldehyde. Live GFP+ transfected cells were identified based on a viability dye staining (Aqua vivid; Invitrogen). Samples were acquired on an LSR II cytometer (BD Biosciences), and data analysis was performed using FlowJo v10.5.3 (Tree Star). An example of the flow cytometry gating strategy is shown in Supplementary Fig S10.

### Protein expression and purification
Expi293F GnTI⁻ cells (Thermo Fisher Scientific, Catalog# A39240) at a density of $1 \times 10^6$ cells/mL were co-transfected with plasmids encoding the Twin-Strep tagged stabilized T/F100 R6. SOSIP.664 with or without LMHS mutations, and furin (DNA ratio 4:1) using polyethylenimine (PEI) or EndoFectin™ Max (GeneCopoeia). One-week post-transfection, the 0.22 μm filtered supernatant was loaded to streptactin XT

resin (IBA Lifesciences) followed by size-exclusion chromatography (SEC) using a Superdex 200 increase 10/300 column (Cytiva) equilibrated with 1x phosphate buffered saline (PBS). The SOSIP.664 trimer peak was harvested as described[63] and protein purity was confirmed by SDS-PAGE.

The expression plasmids encoding the heavy and light chains of 8ANC195 (expression plasmids kindly provided by Michel Nussenzweig, the Rockefeller University) were transiently transfected into Expi293F cells (Thermo Fisher Scientific, Catalog# A14528) using ExpiFectamine 293 transfection as described (Thermo Fisher Scientific). 6-days post-transfection, antibody was purified on Protein A resin from clarified cell supernatant (Thermo Fisher Scientific). Large-scale production of 10–1074 was conducted at Scripps Research Center for Antibody Development & Production (La Jolla, CA) using plasmid DNA for transient transfections of IgG1 10–1074 in 293F cells. Expressed antibody in cell culture supernatant was purified over protein G-Sepharose in phosphate-buffered saline (PBS) (pH 72), eluted with 0.1 M acetic acid (pH 2.8), and dialyzed back into PBS (pH 7.2). The purity of the preparations was checked by size exclusion chromatography on a Superdex 200 10/300 column and determined to be >99% pure. Fabs were generated from an overnight papain digestion of IgG at 37 °C using immobilized papain agarose (Thermo Fisher Scientific). The resulting Fab was separated from Fc and undigested IgG by passage over protein A resin. Fab was further purified by gel filtration using a Superose 6 10/300 column (Cytiva) before use in cryo-EM sample preparation.

## Negative staining and cryo-EM sample preparation, and data collection

T/F100 SOSIP.664 (GnT1⁻ produced) and its LMHS mutant were incubated with 20-fold molar excess of 8ANC195 and 10–1074 Fabs in the presence/absence of 0.5–1 mM temsavir overnight at 4 °C before the purification on a Superose 6 300/10 GL column (Cytiva). The complex peak was harvested, concentrated to 0.8–1.2 mg/mL in 1x PBS buffer. For negative staining, complex at concentration of 5 ug/ml was dispersed on the grid (Formvar/Carbon Square Mesh - Cu, 400 Mesh, UA, FCF400-CU-50) with contrasting agent (2% uranium) and images were taken on the ransmission Electron Microscopes (TEM), Joel1011.

For cryo-EM data collection, 3 μL of protein was deposited on holey carbon copper grids (QUANTIFOIL R 1.2/1.3-2 nm, 300 mesh, EMS) which had been glow-discharged for 20 s at 15 mA using PELCO easiGlow (TedPella Inc). All grids were vitrified in liquid ethane using Vitrobot Mark IV (Thermo Fisher Scientific) with a blot time of 2.5–4 s at 4 °C and 95% humidity. The frozen grids were screened on a FEI Talos Arctica microscope at 200 kV equipped with a FEI Falcon3EC detector using the EPU software (Thermo Fisher Scientific). Cryo-EM data of LMHS/temsavir were acquired on a FEI Glacios electron microscope operating at 200 kV, equipped with a Gatan K3 direct electron detector. Micrographs were collected at a magnification of 45,000 corresponding to a calibrated pixel size of 0.8893 Å, with a total exposure dose of 58 e⁻/ Å². Wild type apo and LMHS apo were acquired on a FEI Titan Krios electron microscope operating at 300 kV equipped with Gatan Bioquantum Image filter-K3 direct electron detector (Gatan Inc) at 20 eV energy slit. Micrographs were collected at a magnification of 105,000 corresponding to a calibrated pixel size of 0.83 Å, with a total exposure dose of 54 e⁻/ Å².

## Cryo-EM data processing, model building and analysis

Motion correction, CTF estimation, particle picking, curation and extraction, 2D classification, ab initio model reconstruction, 3D refinements and local resolution estimation were carried out in cryoSPARC[66,67] or CisTEM[68]. The initial model for T/F100-SOSIP.664-8ANC195 complex[41] (PDB: 6NQD) and 10–1074 Fab[55] (PDB: 4FQ2) were used as modeling templates. The LMHS apo dataset was processed and refined using cisTEM[68]. Local resolution was calculated using Resmap in RELION[69].

Automated and manual model refinements were iteratively carried out in CCP-EM[70], Phenix[71] (real-space refinement) and Coot[72]. Geometry validation and structure quality evaluation were performed by EM-Ringer[73] and Molprobity[74]. Model-to-map fitting cross correlation and figures generation were carried out in UCSF Chimera, Chimera X[75] and PyMOL (The PyMOL Molecular Graphics System, Version 1.2r3pre, Schrödinger, LLC). The cryo-EM data processing workflow is shown in Supplementary Figs. S2–S4 and S6 and statistics of data collection, reconstruction and refinement are described in Supplementary Table S1. The epitope interface analysis was performed in PISA[76].

## Quantification and statistical analysis

Statistics were analyzed using GraphPad Prism version 9.4.1 (GraphPad). Every data set was tested for statistical normality using the Shapiro–Wilk test and this information was used to apply the appropriate (parametric or nonparametric) statistical test. $P$ values < 0.05 were considered significant.

## Reporting summary

Further information on research design is available in the Nature Portfolio Reporting Summary linked to this article.

## Data availability

Cryo-EM reconstructions and atomic models generated during this study are deposited in the Protein Data Bank (PDB) and the Electron Microscopy Data Bank (EMDB) under the following accession codes: PDB: 8DOK, 8G6U, 8CZZ, and 8TTW and EMDB: EMD-27596, EMD-29783, EMD-27103 and EMD-41613. Structural data collection, refinement statistics and codes for deposited structures are provided in the Supplementary Information (Table S1) and Source data for Figs. 1, 2 and S1 are provided with this paper. Source data are provided with this paper.

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

## Acknowledgements

The authors thank the CRCHUM BSL3 and Flow Cytometry Platforms for technical assistance. We thank Agnes L. Chenine (U.S. Military HIV Research Program) for providing the 92TH023 and CM244 Env expressors. We would like to thank Drs. Di Wu and Grzegorz Piszczek from the Biophysics Core Facility, National Heart, Lung, and Blood Institute NIH, Bethesda, MD, USA for help in initial characterization of temsavir complexes using single-molecule mass photometry. We greatly thank Allison R. Zeher, Abraham J. Morton, Zabrina C. Lang from National Cancer Institute and NIH IRP CryoEM Consortium (NICE) for electron microscopy data collection support. Funding for this study was provided by the National Institute of Health grants (R01 AI148379 and R01 AI150322 to A.F.; R01 AI129769 and R01AI174908 to M.P. and A.F.; and R01 AI129801 to A. H.). Support for this work was also provided by P01 GM56550/AI150471 to A.F. and P01 AI162242 to M.P. and Georgia Tomaras. This work was also supported by CIHR foundation grant 352417 to A.F. and Canada Foundation for Innovation (CFI) grant 41027 (to A.F.), and to F.Z. and D.M. by the Division of Intramural Research of the *Eunice Kennedy Shriver* National Institute of Child Health and Human Development, NIH (grant NICHD intramural projects Z1A HD008998). This work was partially supported by the ViiV Healthcare grant 219712 to M.P. A.F. is the recipient of Canada Research Chair on Retroviral Entry RCHS0235 950–232424. J.P. is the recipient of a CIHR doctoral fellowship. R.G. was supported by a MITACS Accélération postdoctoral fellowship. CryoEM facilties at IBBR are supported by the University of Maryland Strategic partnership (MPower). Funding bodies had no role in the design, collection, analysis, or interpretation of the data. The funders had no role in study design, data collection and analysis, decision to publish, or preparation of the manuscript and the contents of this publication are solely the responsibility of the authors.

## Author contributions

J.P., Y.C., A.F. and M.P. designed the study, performed research and analyzed the data; Y.C., W.D.T., F.Z., R.K.H. E.P. and D.M. collected cryo-EM data; Y.C. W.D.T. F.Z., R.K.H. and E.P. processed cryo-EM data and Y.C and W.D.T. solved and refined structures; D.N.N. and S.G. provided BG505 SOSIP.664 preparations for Cryo-EM experiments, R.G., H.M. and M.N. helped with the viral neutralization experiments; A.J.H and V. B. R. provided unique reagents; J.P., Y.C., W.D.T., A.F. and M.P. wrote the manuscript and all authors provided comments or revisions.

## Competing interests

J.P., A.F., and M.P., are inventors on U.S. patent application No. 17/762,333 related to compositions and methods based on HIV gp120 LMHS mutants, in which there are no restrictions on the publication of this manuscript data. The remaining authors declare no competing interests. The views expressed in this manuscript are those of the authors and do not reflect the official policy or position of the Uniformed Services University, US Army, the Department of Defense, or the US Government.
