## [Peer Review File · Nature Communications]

Structure-function Analyses Reveal Key Molecular Determinants of HIV-1 CRF01_AE Resistance to the Entry Inhibitor temsavirREVIEWER COMMENTS

Reviewer #1 (Remarks to the Author):

The article titled "Structure-function Analyses Reveal Key Molecular Determinants of H1 CRF01_AE Resistance to the Entry Inhibitor temsavir" reports a study of temsavir resistance by Env mutations using pseudovirus assay and three cryo-EM structures. Temsavir binds Env and inhibits Env-CD4 interaction, and temsavir is effective against most resistant HIV variants. This study is focused on the temsavir resistance caused by the amino acid variation at the position 375 either naturally in some HIV subtypes or by mutation. The structures of wild-type apo, LMHS mutant apo and temsavir-bound complex were determined by cryo-EM at respectable resolution between 3.14 – 3.2 Å resolution. Extensive analysis of the difference in temsavir binding to wildtype Env (PDB 5U70) and LMHS mutant is provided. The study has implications in understanding temsavir resistance and has the potential for designing new F43 pocket binders.

Major comments

1. The reference structure of wildtype Env-temsavir complex (PDB 5U70) used in the analysis (Fig 5) was determined by x-ray crystallography. To minimize the impact of two different structure determination techniques on the atom-to-atom interaction analysis, it is meaningful to have the reference structure determined in the same condition. Since the authors have determined the wildtype apo structure and the mutant Env-temsavir structure, it would not be difficult for them to determine the wildtype Env-temsavir complex by cryo-EM using the same experimental conditions, antibody etc. as they did for their three current structures.
2. The conclusion (Fig. 4a) that H375 sidechain has steric hindrance with the phenyl ring of temsavir is speculative because H375 sidechain can switch its orientation and permit the inhibitor binding. The resistance may occur by different mechanisms such as the mutation may impact the dynamic of inhibitor entry and/or Kon/Koff rates. The authors may try to obtain the structure of H375 mutant-temsavir complex to see if the drug binding is unfavorable or if there are other underlying mechanisms associated with the drug resistance.
3. The interactions and buried surface area analyses are just listed (Table S2) or displayed as bar charts (Figs 5 and 6). Those results should be discussed in their relevance in the activities such as drug binding, resistance, etc.

Other comments

Line 45: Ref. 2 only discusses nucleotide drugs, not NNRTIs or NRTIs, unlike that referred to in the text.

Line 95: No specific details on the reanalysis of Pancera et al (ref. 11) data are discussed. Are there any noticeable new insights?

Line 103: How do the authors define Asn375 as uncommon compared to Ser/Thr375?

Line 120: How conserved are His61, Gln105, Val108, Asn474, Ile475, and Lys476 and what are the impacts of the mutation on CD4 interactions and viral fitness? Authors may provide a sequence alignment of the pocket region as a supplementary figure and refer to it in their analysis of inhibitor-protein interactions.

Line 233: Define "twenty-two gp120 residues"

Line 287: "These variables should be considered in the structure-based development of more

potent and broader temsavir-like attachment inhibitors." Such design concepts have been considered in HIV NNRTI and protease inhibitor designs. Authors should refer to those articles.

Line 330: I assume, Glu100i is a typo and should be Glu100. Please confirm or define Glu100i.

Line 332: How is Y100b defined? Also, for consistency, use either three or one letter for amino acids in the manuscript.

A figure showing the structural elements of Env monomer will be helpful to visualize.

The green and yellow colors of superimposed structures can be difficult for many. Contrast colors may be used.

Reviewer #2 (Remarks to the Author):

The current study of Prevost et al. identified key residues that contribute to Temsavir resistance which has been observed in strains like CRF01_AE with His375. Although His375 has been considered as a major cause of the resistance, the current study revealed that another 6 residues located at gp120 inner layer all contributed to the drug resistance by analyzing on the reported IC50s of HIV strains against Temsavir. By comparing the cryoEM structures of CRF01_AE SOSIP, CRF01_AE SOSIP LMHS and LMHS with Temsavir, they concluded that these residues including His375 desensitized HIV Env to Temsavir via adjusting gp120 inner layer mobility, binding pocket structure and close-to-open transition of Env trimer. Novelty and significance of this study reside on the detailed molecular mechanism behind the resistance, which will contribute to drug design and optimizing for new entry inhibitor with broader spectrum.

Here are the comments that may help to improve the current study.

1. In the mutagenesis study shown in Fig 1 and 2, the author evaluated the effect of single-point mutation at 375 and multipoint mutation at the other 6 residues. Can the author elaborate on whether these 6 residues are all required? Is there neutralization data to show the effect of each of them? For example, the mutation I108V may not be necessary, since the difference in Temsavir sensitivity as shown in Fig. 2A looks relatively mild.
2. The cryoEM structure was in the presence of 2 antibodies. Could the antibody binding interfere with Temsavir binding? Could there be any impact on beta20-21 mobility and the overall stability of Env?
3. Will there be impact on 10-1074 and 8ANC195 when bound with Temsavir? In fig. S2 and S3, the density of the Fabs become much weaker after temsavir binding, possibly indicating a loose interaction between Fab and Env. Can the author clarify on it?
4. Minor point: Residue name and number is incorrect in fig. S5A layer 2 and 3.

Reviewer #3 (Remarks to the Author):

Summary

Prévost et al. investigate the mechanism of resistance to temsavir, an HIV attachment inhibitor approved for treatment of adults with multidrug-resistant HIV infection. Previous work by others showed that temsavir can inhibit Env function by either allosteric or competitive mechanisms; it was shown that temsavir binding to Env required a small residue at position 375 in the Phe43 cavity and that temsavir binds to an induced pocket under the β 20-21 loop, distinct from the Phe43 cavity. Here Prévost et al. combined mutagenesis with detailed high-resolution, cryoEM analyses to further investigate temsavir resistance by analyzing the contributions of the residue

size at position 375, as well as six mutations in the inner layers of gp120 that these authors previously identified as co-evolving with His375 in temsavir-resistant CRF01_AE strains. Using data from a prior study that analyzed the susceptibility of a panel of Envs to temsavir, the current authors confirmed that there was an inverse relationship between the size of the residue at 375 and sensitivity to temsavir. They also confirm that six residues in gp120 inner layers 1,2 and 3 are also involved in temsavir resistance, which is shown here in different Env contexts. Follow up high-resolution structure studies involving stabilized CRF01_AE Envs with and without 375 and layer mutations conferring temsavir sensitivity revealed that the layer mutations affect β 20-21 loop mobility to accommodate binding of temsavir. The findings indicate that residue 375 is important but not the sole element responsible for resistance in the CRF01_AE Envs. These studies also show that the binding mode of temsavir to Envs can adjust to differences in residues and conformation differences among HIV-1 strains.

Comments

Overall, the iterative studies presented here build on extensive existing work done by this group and others investigating the antiviral mechanisms of temsavir and CD4-mimetics. The findings confirm and extend prior studies by providing new information at the molecular level of the interaction network between inner layers 2 and 3 and movement of the β 20-21 loop that is important for inhibitor and receptor interactions in the context of an CRF01_AE Env. The conceptual advances are mostly incremental, yet the new data should inform the rational design of inhibitors with improved potency and breadth. The studies are well done, and the manuscript is clearly written. CryoEM data were not reviewed. No significant weaknesses.

Minor suggestions

- Color dots in Fig 2A are not distinct enough.
- Suggest adding some discussion about how the newly identified features of the interaction network might influence Env-receptor interactions and virus entry in different Env contexts.
- Consider commenting on how the SOSIP-stabilizing mutations or chaperone antibodies might affect temsavir binding and if it might be different in native Envs.

Dear Editor,

Reviewer #1 (Remarks to the Author):

The article titled “Structure-function Analyses Reveal Key Molecular Determinants of H1 CRF01_AE Resistance to the Entry Inhibitor temsavir” reports a study of temsavir resistance by Env mutations using pseudovirus assay and three cryo-EM structures. Temsavir binds Env and inhibits Env-CD4 interaction, and temsavir is effective against most resistant HIV variants. This study is focused on the temsavir resistance caused by the amino acid variation at the position 375 either naturally in some HIV subtypes or by mutation. The structures of wild-type apo, LMHS mutant apo and temsavir-bound complex were determined by cryo-EM at respectable resolution between 3.14 – 3.2 Å resolution. Extensive analysis of the difference in temsavir binding to wildtype Env (PDB 5U7O) and LMHS mutant is provided. The study has implications in understanding temsavir resistance and has the potential for designing new F43 pocket binders.

Response: We thank the reviewer for this positive assessment of our work.

Major comments:

1. The reference structure of wildtype Env-temsavir complex (PDB 5U7O) used in the analysis (Fig 5) was determined by x-ray crystallography. To minimize the impact of two different structure determination techniques on the atom-to-atom interaction analysis, it is meaningful to have the reference structure determined in the same condition. Since the authors have determined the wildtype apo structure and the mutant Env-temsavir structure, it would not be difficult for them to determine the wildtype Env-temsavir complex by cryo-EM using the same experimental conditions, antibody etc. as they did for their three current structures.

Response: We thank reviewer for the suggestion of redoing the BG505 complex structure using Cryo-EM and the same chaperone Fabs that we used for the CRF01_AE structures. We solved a 3.0Å resolution Cryo-EM structure of BG505 SOSIP.664 in complex with temsavir and the two chaperone antibodies 8ANC195 and 10-1074. This structure is now included (Table S1, S2; Fig. S6 and S7) and used for structural comparisons such as the angle of opening of the trimer (Fig. 3), the temsavir binding pocket (Fig. 5), and the 10-1074 Fab interface (Fig. S9). As described in the manuscript text, the structural alignments of the two temsavir-BG505 SOSIP.664 complexes determined by us and previously by x-ray crystallography by Pancera *et al.* indicate a close similarity of the overall trimer assembly (root mean square deviation (RMSD) of 1.23Å and 1.19Å for the trimer and protomer respectively) as well as the temsavir binding pocket. In both structures, the temsavir binding pocket is formed by the same set of residues that contribute similar BSA to the pocket. A major difference is observed in the regions accommodating the phenyl ring of temsavir with noticeable displacement of the ring. In the complex from this study, the phenyl ring is shifted approximately 1Å up to pack against Phe382 which could be in response of the slight shift in position of Trp427 relative to temsavir or to the different rotamer observed for the Phe382 phenyl ring in this structure. Most of the other side chains within the pocket are largely superimposable highlighting the similarity of the two temsavir binding pockets.

2. The conclusion (Fig. 4a) that H375 sidechain has steric hindrance with the phenyl ring of temsavir is speculative because H375 sidechain can switch its orientation and permit the inhibitor binding. The resistance may occur by different mechanisms such as the mutation may impact the dynamic of inhibitor entry and/or Kon/Koff rates. The authors may try to obtain the structure of H375 mutant-temsavir complex to see if the drug binding is unfavorable or if there are other underlying mechanisms associated with the drug resistance.

Response: The presence of His375 presumably blocks the binding site and we essentially see no binding of temsavir to CRF01_AE wild-type, or temsavir-sensitive strains that have an introduced His375 (e.g. JR-FL, YU2) (See Figs 1d, 2c, 2e, 2g). The lack of efficient binding prevents us from getting a temsavir bound structure with any Env that has His375. For example, we did attempt structural studies using the CRF01_AE wild-type and CRF01_AE LM His375 variants in complex with temsavir (similar to what was obtained CRF01_AE LMHS variant), but after determining the structures of such ‘complexes’ found that temsavir was not bound (there was no density for the compound). Previous study by *Schader et al.* (PMID: 22615295) performed molecular docking simulation of temsavir-like molecules on CRF01_AE Env which suggested that the His375 sidechain prevent the insertion of the drug phenyl ring within the Phe43 cavity, forcing it to use an unfavorable binding mode.

3. The interactions and buried surface area analyses are just listed (Table S2) or displayed as bar charts (Figs 5 and 6). Those results should be discussed in their relevance in the activities such as drug binding, resistance, etc.

Response: The buried surface area analysis (Table S2) and bar charts in Fig 6 are mainly discussed in the section regarding the chaperone antibody interfaces for 8ANC195 and 10-1074 (in the last results subsection). They are used as a tool for comparison of the antibody-Env interface. To discuss relevance of pocked residues in drug resistance we modified Fig. 5 to add Weblogo depiction of sequence conservation for temsavir and used this new information along with the BSA bar chart to discuss this issue in manuscript lines 306-322:

‘Despite the observed conformational differences of temsavir within the T/F100 LMHS SOSIP.664 and BG505 SOSIP.664 binding cavities, the Env residues contributing to temsavir binding are highly conserved. Using residue BSA to stratify residue contribution to the interface, temsavir mostly relies on interactions to Env residues that are strictly conserved in over 99% of HIV-1 sequences (residues Trp112, Leu116, Phe382, and Trp427, Fig. 5c), highly conserved with approximately 97% or more of HIV-1 sequences (residues Asp113, Lys117, and Val255) or conserved with limited sequence variation (residues Thr/Lys202, Ile/Val424, Met/Arg/Leu426, Lys/Gln/Arg432 and Met/Ile475). The high degree of conservation implies that Env is only able to tolerate limited sequence diversity at these positions. Temsavir can accommodate many of these changes. The Thr202Lys change can be accommodated by a rearrangement of the methyltriazole ring in temsavir as mentioned previously. Asp113 which is involved in one of the two hydrogen bonds with temsavir is rarely any other residue, but the infrequently seen Glu113 or Asn113 can also form a hydrogen bond with temsavir. And position 424 contributes to the binding pocket through van der Waals interactions that can be made equally well by different residues (i.e. Ile or

Val). Other key interactions with temsavir are dependent on main chain atoms such as Lys/Gln/Arg432 or the hydrogen bond between Trp427 and temsavir. One of the few temsavir pocket residue sequence changes that can influence temsavir binding is His at position 375. Temsavir resistance in many other cases is likely to come from conformational changes induced by residue changes outside of the binding pocket.'

Other comments

Line 45: Ref. 2 only discusses nucleotide drugs, not NNRTIs or NRTIs, unlike that referred to in the text.

Response: We thank the reviewer for pointing that out. We have now added references to account for both classes of reverse transcriptase inhibitors (NRTI and NNRTI).

Line 95: No specific details on the reanalysis of Pancera et al (ref. 11) data are discussed. Are there any noticeable new insights?

Response: In the original article by *Pancera et al.*, the neutralization data is mainly used to evaluate the overall breadth of Temsavir against a panel of 208 HIV-1 Env. We took advantage of this available datasets to dig further and reanalyzed the data to pinpoint the subtle differences in temsavir sensitivity between HIV-1 clades (Fig 1a). We also focused on the contribution of specific polymorphic residues (LMHS) to this phenotype (Fig. 1b and 2a). This is now clarified at lines 96-97.

Line 103: How do the authors define Asn375 as uncommon compared to Ser/Thr375?

Response: We have added a Weblogo depiction of sequence conservation for temsavir pocket residues to Fig. 5. This was generated from all clades of HIV-1 Env sequences in the Los Alamos HIV database. At position 375, Ser is the most common residue followed by His and Thr. Asn occurs rarely and is not visible in this plot. Asn occurs at most in approximately 2 to 3% of all sequences in this database versus approximately 80% for Ser and approximately 10% for Thr.

Line 120: How conserved are His61, Gln105, Val108, Asn474, Ile475, and Lys476 and what are the impacts of the mutation on CD4 interactions and viral fitness?

Response: The degree of conservation of these residues has been previously reported by our group (PMID: 27928014, 32457241). His61, Gln105, Val108, Asn474, Ile475, and Lys476 are specifically enriched in CRF01_AE isolates (>75%) compared to other HIV-1 major clades. These previous studies have demonstrated that His375 and LM changes have coevolved in CRF01_AE strains to secure CD4 binding and obtain optimal infectivity. This has been clarified at lines 123-124.

Authors may provide a sequence alignment of the pocket region as a supplementary figure and refer to it in their analysis of inhibitor-protein interactions.

Response: We thank the reviewer for this suggestion. We have added a Weblogo depiction of sequence conservation for temsavir pocket residues to Fig. 5c and have modified the text accordingly (lines 306-322).

Line 233: Define “twenty-two gp120 residues”

Response: We apologize for the ambiguity. These are the twenty-two (twenty-four in the new structure) residues that line the temsavir binding pocket. These residues are now listed in Fig. 5c.

Line 287: “These variables should be considered in the structure-based development of more potent and broader temsavir-like attachment inhibitors.” Such design concepts have been considered in HIV NNRTI and protease inhibitor designs. Authors should refer to those articles.

Response: We have added appropriate references regarding the structure-based design of protease, reverse transcriptase and integrase inhibitors, as well as other classes of Env antagonists. The text was modified at lines 305-306.

Line 330: I assume, Glu100i is a typo and should be Glu100. Please confirm or define Glu100i.

Response: Our apologies. Glu100i should be Glu100I. Residues are numbered with Kabat antibody numbering where insertions to the CDRH3 are denoted with letters starting at position 100. 10-1074 has a long CDRH3 with an insertion spanning residues 100A to 100P. A reference to the Kabat numbering system has been added to the text.

Line 332: How is Y100b defined? Also, for consistency, use either three or one letter for amino acids in the manuscript.

Response: Thank you for noticing this. Y100b should be Tyr100B. The 100B comes from the Kabat numbering as mentioned previously. We have switched to the three-letter code for amino acids in the text.

A figure showing the structural elements of Env monomer will be helpful to visualize.

Response: We thank the reviewer for this comment and have added a panel showing the secondary elements for the trimer in high contrast (Fig. 4a). Due to space limitation only those secondary elements near the binding pocket are labelled.

The green and yellow colors of superimposed structures can be difficult for many. Contrast colors may be used.

Response: We have added a high contrast view in a new panel with secondary structure elements to help in the interpretation of the superimposed structures.

Reviewer #2 (Remarks to the Author):

The current study of Prevost et al. identified key residues that contribute to Temsavir resistance which has been observed in strains like CRF01_AE with His375. Although His375 has been considered as a major cause of the resistance, the current study revealed that another 6 residues located at gp120 inner layer all contributed to the drug resistance by analyzing on the reported IC50s of HIV strains against Temsavir. By comparing the cryoEM structures of CRF01_AE SOSIP, CRF01_AE SOSIP LMHS and LMHS with Temsavir, they concluded that these residues including His375 desensitized HIV Env to Temsavir via adjusting gp120 inner layer mobility, binding pocket structure and close-to-open transition of Env trimer. Novelty and significance of this study reside on the detailed molecular mechanism behind the resistance, which will contribute to drug design and optimizing for new entry inhibitor with broader spectrum.

Response: We thank the reviewer for the positive assessment of our manuscript.

Here are the comments that may help to improve the current study.

1. In the mutagenesis study shown in Fig 1 and 2, the author evaluated the effect of single-point mutation at 375 and multipoint mutation at the other 6 residues. Can the author elaborate on whether these 6 residues are all required? Is there neutralization data to show the effect of each of them? For example, the mutation I108V may not be necessary, since the difference in Temsavir sensitivity as shown in Fig. 2A looks relatively mild.

Response: We thank the reviewer for this interesting suggestion. We have generated new mutants in Env_{YU2} to evaluate the key contribution of each individual inner domain layer (Layer 1: Y61H, Layer 2: H105Q/I108V and Layer 3: D474N/M475I/R476K) to temsavir neutralization sensitivity. It appears that layer 3 mutations contribute more to temsavir resistance in this HIV-1 Env backbone. The results are shown in Fig. S1 and discussed in the text at lines 133-134.

2. The cryo-EM structure was in the presence of 2 antibodies. Could the antibody binding interfere with Temsavir binding? Could there be any impact on beta20-21 mobility and the overall stability of Env?

Response: We solved a cryo-EM structure of BG505 SOSIP.664 bound to temsavir using the same chaperone antibodies used for the CRF01_AE SOSIP structures and added to the manuscript. This in combination with the temsavir bound BG505 SOSIP.664 from Pancera *et al.* with a different set of chaperone antibodies gives us a third example of a temsavir bound Env structure. Based on these structures, there is no evidence that the chaperone antibodies are inhibiting temsavir binding. These two chaperone antibodies were purposely selected for their far-distant epitope from the temsavir binding pocket and their compatibility with the State 1 “closed” Env conformation, the same conformation preferred by temsavir.

3. Will there be impact on 10-1074 and 8ANC195 when bound with Temsavir? In fig. S2 and S3, the density of the Fabs become much weaker after temsavir binding, possibly indicating a loose interaction between Fab and Env. Can the author clarify on it?

Response: The local resolution plots in the supplemental figures do show different degrees of movement for the 10-1074 and 8ANC195 Fabs between the different structures. This is most pronounced for the constant domains of the Fabs, but to a lesser extent is also true for the variable domains of the Fab. In all the structures, the regions of the Fabs that interacted directly with the trimer were the best defined and had the highest resolution density estimates. But it is hard to differentiate the part of this that is due to temsavir binding from the part that is due to data quality. The number of particles collected and how well they align in making the 3-dimensional reconstruction also contribute to disorder in these regions. We can say that the antibody trimer interfaces were similar in all the structures. Temsavir may have had an effect on glycan mobility which is most evident in the 10-1074 epitope since it mainly relies on N332, but the differences in the data prevent us from making any strong conclusions about this. In this regard, a previous study has shown that temsavir does not affect 10-1074 binding, even at extremely high concentrations (PMID: 30971821).

4. Minor point: Residue name and number is incorrect in fig. S5A layer 2 and 3.

Response: We thank the reviewer for noticing this and have corrected the figure.

Reviewer #3 (Remarks to the Author):

Summary

Prévost et al. investigate the mechanism of resistance to temsavir, an HIV attachment inhibitor approved for treatment of adults with multidrug-resistant HIV infection. Previous work by others showed that temsavir can inhibit Env function by either allosteric or competitive mechanisms; it was shown that temsavir binding to Env required a small residue at position 375 in the Phe43 cavity and that temsavir binds to an induced pocket under the β 20-21 loop, distinct from the Phe43 cavity. Here Prévost et al. combined mutagenesis with detailed high-resolution, cryoEM analyses to further investigate temsavir resistance by analyzing the contributions of the residue size at position 375, as well as six mutations in the inner layers of gp120 that these authors previously identified as co-evolving with His375 in temsavir-resistant CRF01_AE strains. Using data from a prior study that analyzed the susceptibility of a panel of Envs to temsavir, the current authors confirmed that there was an inverse relationship between the size of the residue at 375 and sensitivity to temsavir. They also confirm that six residues in gp120 inner layers 1,2 and 3 are also involved in temsavir resistance, which is shown here in different Env contexts. Follow up high-resolution structure studies involving stabilized CRF01_AE Envs with and without 375 and layer mutations conferring temsavir sensitivity revealed that the layer mutations affect β 20-21 loop mobility to accommodate binding of temsavir. The findings indicate that residue 375 is important but not the sole element responsible for resistance in the CRF01_AE Envs. These studies also show that the binding mode of temsavir to Envs can adjust to differences in residues and conformation differences among HIV-1 strains.

Comments

Overall, the iterative studies presented here build on extensive existing work done by this group

and others investigating the antiviral mechanisms of temsavir and CD4-mimetics. The findings confirm and extend prior studies by providing new information at the molecular level of the interaction network between inner layers 2 and 3 and movement of the β 20-21 loop that is important for inhibitor and receptor interactions in the context of an CRF01_AE Env. The conceptual advances are mostly incremental, yet the new data should inform the rational design of inhibitors with improved potency and breadth. The studies are well done, and the manuscript is clearly written. CryoEM data were not reviewed. No significant weaknesses.

Response: We thank the reviewer for the positive assessment of our manuscript.

Minor suggestions

-Color dots in Fig 2A are not distinct enough.

Response: Color dots for Fig 1b and Fig 2a were adjusted to be more distinctive.

-Suggest adding some discussion about how the newly identified features of the interaction network might influence Env-receptor interactions and virus entry in different Env contexts.

Response: The effect of the LMHS residue network on CD4 interaction and virus entry has already been extensively dissected by our group in key studies by [Zoubchenok *et al.* J Virol 2017] and [Prévost *et al.* mBio 2020]. These studies showed that the presence of His375 in CRF01_AE strains is sufficient to secure CD4 binding and mediate viral entry, while other HIV-1 major clades (A, B, C) displaying a smaller residue at position 375 (Ser375 or Thr375) require the presence of compensating inner domain layer mutations (LM) to secure CD4 binding and mediate viral entry. This is now mentioned at lines 123-124.

-Consider commenting on how the SOSIP-stabilizing mutations or chaperone antibodies might affect temsavir binding and if it might be different in native Envs.

Response: We have added to the text a structure of BG505 SOSIP with temsavir bound and the same set of chaperone antibodies that was used for the CRF01_AE temsavir structure. Based on these structures there is no evidence that the chaperone antibodies are inhibiting temsavir binding. However, we only see the final complex. Effects on temsavir affinity, beta20-21 mobility, or stability are probably best assessed by other methods. Likewise, although the SOSIP-stabilizing mutations may have an indirect effect on temsavir binding, cryo-EM might not be the best way to assess this. Other techniques might be better suited to answer this question. We are limited by the fact that we need higher resolution data to see temsavir in the structure. Using SOSIP-stabilized trimers aids in this. While it would be nice to have a structure of a temsavir bound native membrane associated trimer at the same resolution, we are not in the position to do this at this time.

REVIEWERS' COMMENTS

Reviewer #1 (Remarks to the Author):

The authors have addressed all reviewers' comments and suggestions appropriately by conducting additional experiments and analyses. I recommend publishing the manuscript in *Nature Communications*, however, I have two minor suggestions.

1. The reference 44 may not be the most appropriate one describing conformational flexibility in NNRTI design. I would suggest using DOI:10.1021/jm030558s and/or DOI:10.1021/jm040840e that were first to describe how the conformational flexibility helps overcome drug resistance for the drugs rilpivirine and etravirine.
2. At couple places (lines 694 and 800) in the manuscript, the experimental cryo-EM density is referred as "electron density". The electron density is obtained from diffraction data. A cryo-EM density map does not reflect the density of electrons at the atomic positions. Please refer the maps as experimental density map or cryo-EM density map.

Reviewer #2 (Remarks to the Author):

The author has addressed all my questions adequately in the revision. I have no further question to the author.

Reviewer #1 (Remarks to the Author):

The authors have addressed all reviewers' comments and suggestions appropriately by conducting additional experiments and analyses. I recommend publishing the manuscript in Nature Communications, however, I have two minor suggestions.

Response: We thank the reviewer for the positive assessment of our manuscript and his/her recommendation to publish our manuscript in Nat. Comm.

1. The reference 44 may not be the most appropriate one describing conformational flexibility in NNRTI design. I would suggest using DOI:10.1021/jm030558s and/or DOI:10.1021/jm040840e that were first to describe how the conformational flexibility helps overcome drug resistance for the drugs rilpivirine and etravirine.

Response: We agree with the reviewer that reference 44 is not the best to talk about conformation restraints in NNRTI design. We have modified the reference to NNRTI to those suggested by reviewer (now ref 44 and 45).

2. At couple places (lines 694 and 800) in the manuscript, the experimental cryo-EM density is referred as "electron density". The electron density is obtained from diffraction data. A cryo-EM density map does not reflect the density of electrons at the atomic positions. Please refer the maps as experimental density map or cryo-EM density map.

Response: In the first instance (originally line 694 and currently line 879 in the Figure 5 legend) we have modified electron density to be cryo-EM density. In the second instance (originally line 800 and currently Figure S7 legend in the Supplementary Figures file) we have modified electron density maps to be experimental density maps since the first map is a cryo-EM density map and the second map is an electron density map determined by x-ray crystallography.

Reviewer #2 (Remarks to the Author):

The author has addressed all my questions adequately in the revision. I have no further question to the author.

Response: We thank the Reviewer for his/her response